# Systematic computational assessment of atrial function impairment due to fibrotic remodeling in electromechanical properties

Åshild Telle[1,2], Ahmad Kassar[3], Nadia Chamoun[3], Romanos Haykal[3], Alejandro Gonzalo[4], Tori Hensley[3], Yaacoub Chahine[3], Oscar Flores[5], Juan C. del Álamo[1,3,6], Nazem Akoum[3], Christoph M. Augustin[7,8‡*], Patrick M. Boyle[1,2,3,6,9‡*]

**1** Department of Bioengineering, University of Washington, Seattle, Washington, United States of America, **2** eScience Institute, University of Washington, Seattle, Washington, United States of America, **3** Division of Cardiology, Department of Medicine, University of Washington, Seattle, Washington, United States of America, **4** Department of Mechanical Engineering, University of Washington, Seattle, Washington, United States of America, **5** Department of Aerospace Engineering, Universidad Carlos III de Madrid, Leganés, Spain, **6** Center for Cardiovascular Biology, University of Washington, Seattle, Washington, United States of America, **7** Gottfried Schatz Research Center for Cell Signaling, Metabolism and Aging – Division of Medical Physics and Biophysics, Medical University of Graz, Graz, Austria, **8** BioTechMed-Graz, Graz, Austria, **9** Institute for Stem Cell and Regenerative Medicine, University of Washington, Seattle, Washington, United States of America

‡ Joint senior authors.
* christoph.augustin@medunigraz.at (CMA); pmjboyle@uw.edu (PMB)

## Abstract

Cardiac fibrosis is a pathological condition associated with many cardiovascular diseases. Atrial fibrosis leads to reduced atrial function, resulting in diminished blood flow and an increased risk of stroke. This reduced function arises from altered myocardial electrophysiological and mechanical properties. Identifying the relative importance of these fibrosis-associated properties can reveal the most significant determinants of left atrial function impairment. In this study, we used a computational framework to investigate the relative importance of various fibrosis-associated properties. Our model, a 3D electromechanical framework coupled with a 0D circulatory model, incorporated patient-specific geometries and fibrosis distributions from clinical imaging data. Nine parameters related to fibrotic remodeling (conduction velocity, ion channel expression levels, cell- and tissue-scale contractility, and stiffness) were analyzed using two sensitivity analysis schemes: a one-factor-at-a-time setup, allowing for analysis of isolated effects, and a fractional factorial design, enabling examination of combined effects. As output, we tracked various metrics derived from model-predicted pressure-volume loops. Impairment of L-type calcium current ($I_{CaL}$) was most detrimental (up to 64% reduction in A-loop area of the left atrial pressure-volume relationship, quantifying work performed during atrial contraction). Conversely, reduced inward rectifier current ($I_{K1}$) led to improved atrial function (up to 27% increase in A-loop area). Detailed analysis of spatiotemporal distributions linked these effects to changes in intracellular calcium handling. Fractional factorial design

**Data availability statement:** The organ-level simulations were performed using proprietary software (CARPentry, Numericor; https://numericor.at), which cannot be made publicly available. The related openCARP software (https://opencarp.org/) is freely available for non-commercial use (https://opencarp.org). Supplementary Data for this study are published in a Dryad Repository (https://doi.org/10.5061/dryad.g79cnp62q), including: geometries used in the simulations, for all three patients and both fibrosis levels; the script used to calibrate CV values for all fibrotic combinations; all parameter combinations used in the two sensitivity analysis setups (OFAT and FFD); scripts used to perform all simulations with CARPentry, including both unloading and sensitivity analysis scripts; pressure and volume data from all cycles from all of the above-mentioned simulations for all three patients and both fibrosis levels. The code for performing the post-simulation sensitivity analysis is publicly available via Zenodo (https://doi.org/10.5281/zenodo.15693590), as is the code for material parameter estimation (https://doi.org/10.5281/zenodo.15693906).

**Funding:** This study was supported by the US National Institutes of Health grants R01-HL158667 (NA, JCA, CMA, PMB) and R01-HL160024 (JCA), and by the Austrian Science Fund grant 10.55776/P37063 (CMA). The funders did not play any role in the study design, data collection and analysis, decision to publish, or preparation of the manuscript.

**Competing interests:** The authors have declared that no competing interests exist.

analysis revealed that combination with other parameter changes blunted the impact of reduced $I_{CaL}$ but amplified the impact of reduced $I_{K1}$. Future research focusing on $I_{K1}$ and $I_{CaL}$ could be highly significant for clinical and scientific advances. Modeling work can help evaluate left atrial function among larger patient cohorts, focusing on strain analysis. Our work could also be extended to spatiotemporal simulations of blood flow and thrombosis, shedding light onto the mechanisms underlying atriogenic stroke.

## Author summary

Cardiac fibrosis is a process where healthy heart muscle is replaced with non-conductive, non-contractile tissue. This change disrupts how the heart beats and contracts. In the left atrium, fibrosis is linked to atrial fibrillation and a higher risk of stroke, the latter due to impaired pumping and altered blood flow. In this study, we used a detailed computer model of the heart, based on real patient-specific left atrial geometries and fibrosis patterns, to understand how different fibrosis-related changes affect atrial function. We tested nine features of the heart's electrical and mechanical behavior that are known to change during fibrosis, aiming to identify which ones have the most impact on the atrial function. We found that reducing the L-type calcium current — an important signal for muscle contraction — caused the greatest decrease in atrial performance. Surprisingly, reducing the inward rectifier potassium current improved it. These effects were tied to changes in calcium handling inside heart cells. Our findings highlight promising directions for future heart disease research and treatment.

## 1 Introduction

Cardiac fibrosis is prevalent in cardiovascular disease and contributes to left atrial (LA) dysfunction. LA fibrosis is strongly associated with atrial fibrillation (AF) and ischemic stroke [1–3]. Fibrotic remodeling encompasses a series of complex pathological events involving myocyte death, expansion of the extracellular matrix, and subcellular electromechanical changes [4,5]. These alterations reduces LA function, in which relative reduction can be quantified to support mechanistic insight and clinical risk stratification.

Fibrotic remodeling profoundly impacts myocardial electrophysiological (EP) and mechanical properties. Structural tissue-level changes (myocyte necrosis) lead to decreased conduction velocity (CV) [6] and reduced myocardial force generation. Subcellular remodeling reduces ion channel conductances [1,5], also leading to reduced contractility. LA fibrosis is furthermore linked to increased atrial stiffness [7], often attributed to up-regulated collagen crosslinking [8] and changes in collagen composition [9], likely combined with myocyte stiffening [10–12]. These changes affect the cardiac function, but our understanding of their relative contributions remains limited.

Computational modeling offers a powerful approach to elucidate the consequences of fibrosis-related alterations. By tuning parameters in physiologically informed models, one can predict consequences of specific pathological changes. Computational EP models of fibrotic LA have been used to study the connection between fibrosis and AF [13–16], revealing how altered CV and ion channel expression in fibrotic regions influence arrhythmia inducibility and spatial characteristics. Multi-physics, multi-scale modeling frameworks have been used to assess the impact of LA remodeling (including fibrosis) [17] and AF (without fibrosis) [18,19] on pressure-volume (PV) relationships. Hemodynamic effects of fibrotic remodeling have also been explored via computational fluid dynamics analyses [20,21]. All of these studies have shown utility in quantifying clinically relevant variables and enabling targeted investigations.

There are several approaches to make use of computational models to investigate individual impact of various parameters. One-factor-at-a-time (OFAT) analysis provides a systematic and straight-forward way to assess the isolated impact of each parameter [22,23], however, this does not account for combined or cooperative effects. Fractional factorial designs (FFD) setups offer a powerful method for reducing the parameter space by examining key combinations in a balanced and efficient manner [24,25]. It allows for investigating individual and interactive parameter effects, while minimizing the number of required experiments. FFDs and design of experiment protocols have previously been used to study parameter sensitivity in ionic EP models [26,27] and computational fluid dynamics analysis [28,29]. While FFD have been used only sparingly in cardiac computational modeling, the methodology hold potential for exploring large parameter spaces in detailed models of cardiac function [30]. To our knowledge, FFD has not yet been applied to multi-physics, organ-scale cardiac models.

This study combines computational modeling with OFAT and FFD analyses to disentangle the effects of various fibrosis-associated properties on atrial function. We performed simulations using three patient-specific LA geometries with corresponding fibrosis distributions, perturbing nine electromechanical parameters in fibrotic regions. First, we conducted an OFAT sensitivity analysis to assess the isolated effect of each parameter. Next, we explored spatiotemporal distributions of membrane potential, intracellular calcium, and active tension from the simulations in which these parameters were changed to elucidate the mechanisms driving their importance. To gain deeper understanding of combined interactive effects, we performed a more detailed sensitivity analysis using a $2^{9-5}$ FFD scheme. We then repeated both analyses with 50% synthetically elevated fibrosis in the same three geometries to assess the effect of an increasing fibrosis burden. As metrics, we tracked A-loop area, booster function, reservoir function, conduit function, and upstroke pressure difference during contraction, all derived from model-predicted PV loops. Sensitivity analyses based on these output values were used to identify the most influential parameters.

## 2 Methods

### 2.1 Ethics statement

The study was approved by the Institutional Review Board of the University of Washington (STUDY00015081). A written statement of consent was obtained from each patient.

### 2.2 Patient recruitment

We obtained three patient-specific LA geometries with corresponding spatial fibrosis and electroanatomical mapping (EAM) data. Participants were recruited at the University of Washington Medical Center, all of whom had AF and were scheduled for ablation. Patient demographics, LA clinical measurements, and modeling parameters are reported in Table 1.

Patients were directly recruited for this and subsequent studies, and not part of earlier cohorts. We included AF patients undergoing ablation procedures, with availability of LGE-MRI suitable for LA wall segmentation and fibrosis quantification, and indicated high-density EAM of the LA performed within a 1-month time-frame of the MRI scan. Exclusion criteria

**Table 1.** **Patient-specific demographics, clinical measurements, and derived parameters.**

| | Patient 1 | Patient 2 | Patient 3 |
|---|---|---|---|
| Age | 81 | 63 | 70 |
| Sex | F | M | F |
| LA end-diastolic volume (mL) | 110.0 | 112.5 | 60.0 |
| Body surface area (m$^2$) | 1.98 | 2.33 | 1.73 |
| LA volume index (mL/m$^2$) | 55.5 | 48.3 | 34.7 |
| LA emptying fraction (LAEF; %) | 39.24 | 51.56 | 58.20 |
| LA fibrosis burden (%)* | 15.6 | 23.9 | 17.9 |
| Synthetically increased fibrosis burden (%)* | 23.4 | 35.85 | 26.85 |
| LA wall thickness (min–max; mm) | 1.4–2.4 | 2.3–3.0 | 1.6–2.4 |
| Total LA activation time (ms) | 82.0 | 56.0 | 116.8 |
| CV longitudinal direction (CV$_L$; m/s) | 1.1500 | 1.5730 | 0.835 |
| CV transverse direction (CV$_T$; m/s) | 0.5142 | 0.7035 | 0.373 |

*of main LA body and appendage (excluding pulmonary veins)

were prior atrial ablation, prior cardiac surgery, contraindications to MRI or gadolinium contrast (e.g., device incompatibility, severe renal dysfunction, pregnancy, gadolinium sensitivity), and inability to obtain diagnostic-quality images due to arrhythmia or habitus constraints. The recruited patients included in this study all satisfied pre-specified data-quality requirements (diagnostic LGE-MRI enabling fibrosis segmentation and EAM with adequate LA surface coverage and point density).

## 2.3 Patient geometries and fibrosis distributions

LA geometries with corresponding fibrosis distributions were obtained from pre-ablation LGE-MRI images. MRI scans were taken at the end of atrial diastole (just prior to contraction). Segmentation, processing, and analysis of raw MRI scans were performed by Merisight (Marrek Inc., Salt Lake City, UT), as previously described [31] and applied [15,16,32]. Fibrosis burdens are reported in Table 1 and annotated in Fig 1. The geometries were represented as 3D triangulated surfaces with fibrosis distributions (LGE maps), which we extruded by 2 mm [33,34] outward to create volumetric geometries using CARPentry Studio (NumeriCor GmbH, Graz, Austria) [35]. We used a uniform thickness following literature rather than personalized values (with ranges reported in Table 1), as we did not have spatial thickness values co-registered with the geometries.

Fibrosis distributions in the volumetric geometries were interpolated from LGE maps. We used a priority-based protocol ensuring that the total fibrotic burdens matched those reported by Merisight. Volumetric mesh elements were sorted and given a priority value according to the interpolated normalized LGE value. Following this order, we assigned elements one by one as fibrotic until the volumetric fibrotic ratio matched the clinically reported fibrosis burden. Fibrotic tissue was assumed to be uniformly distributed from the epicardium to the endocardium, with no transmural variation. Fibrosis burden was calculated as a percentage of mesh elements excluding the pulmonary veins (i.e., only the LA body and appendage), for original LGE maps and our volumetric geometries alike. To explore the effects of extended fibrosis, we applied the same method but with a target fibrosis burden increased by 50% (×1.5). Effectively, this amounted to the identification of a lower cutoff value for normalized LGE in order for a element to be included. In practice, this generally resulted in expansion of fibrotic areas, although not necessarily uniformly since the approach was based on normalized intensity as opposed to geometric region growing. Fig 1A illustrates how fibrotic regions were mapped from LGE maps to volumetric geometries for both levels of fibrosis. Fig 1B displays the resulting volumetric geometries with corresponding fibrosis distributions.

Volumetric geometries were further augmented as needed for our electromechanical simulations. Pulmonary veins are clinically assumed passive post-ablation and were hence not included as LGE map distributions. However, they were

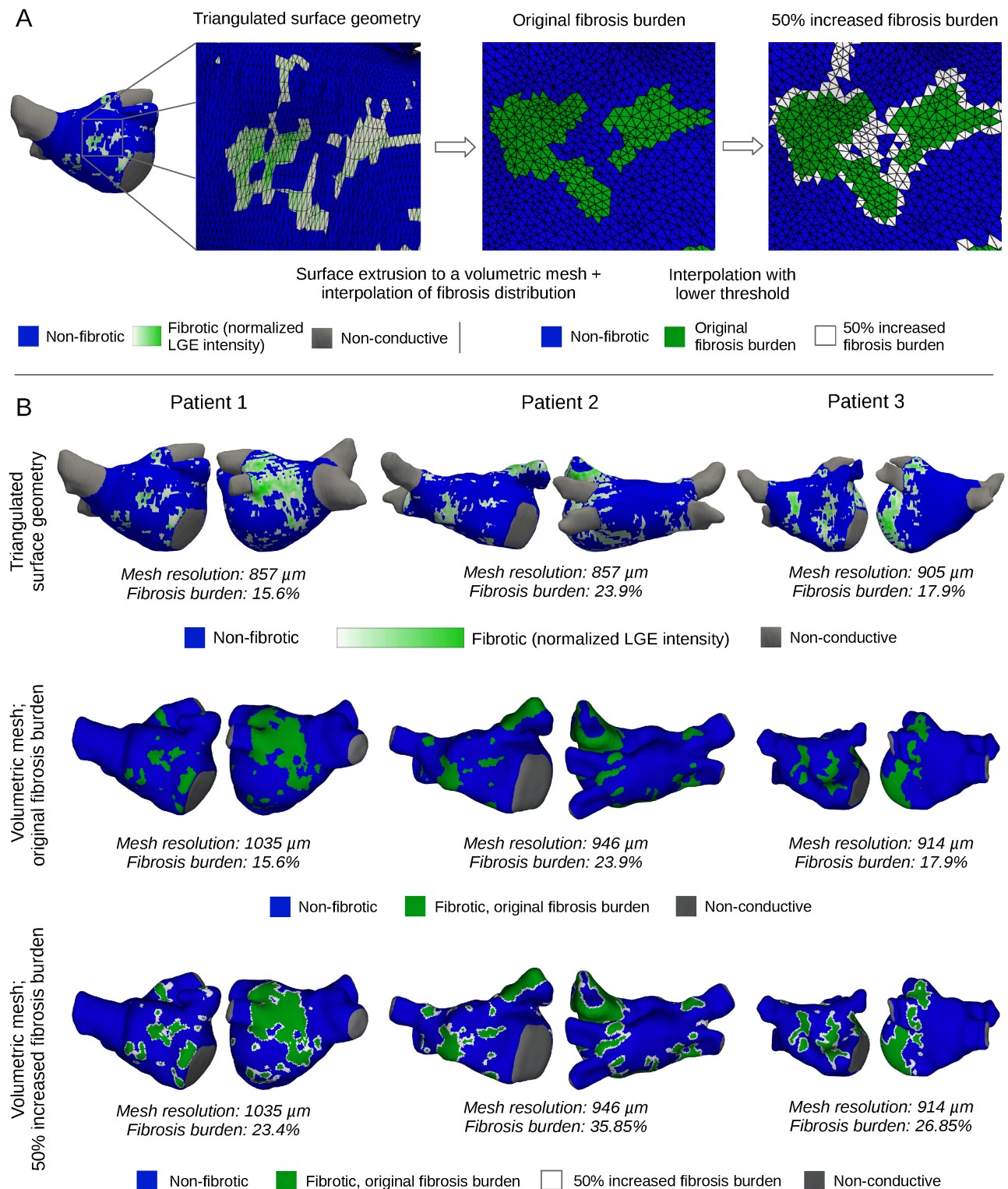

**Fig 1. Patient-specific geometries with fibrosis distributions.** (A) Workflow for mesh generation and fibrosis mapping: Starting with the original triangulated surfaces (left), we generated volumetric geometries. Fibrosis distributions were mapped via interpolation to preserve the original fibrosis burden (middle) and to reach a 50% synthetically elevated fibrosis burden (right). (B) Original shell geometries (top row), volumetric geometries with original fibrosis distributions (middle row), and volumetric geometries with increased fibrosis distributions (bottom row) for the three patient geometries considered in this study. Mesh resolutions (average tetrahedral edge length) and fibrotic burdens are included as annotations.

included in our modeling geometries without corresponding fibrosis distributions. Artificial pulmonary vein caps and a mitral valve representation were added to the geometry to define boundary conditions and preserve anatomical orifice shape even under large deformations. Fiber direction maps (delineating the longitudinal myocyte directions) were generated for each geometry using a rule-based method defined by LA landmarks, similar to those described in previous publications [35,36]. The same fiber architecture was used in both baseline and fibrotic simulations. Volumetric meshes are publicly available in a Dryad Repository [37].

## 2.4 CV calibration and identification of earliest activation locations

EAM was performed and recorded for all patients during the ablation procedure, yielding LA activation maps in sinus rhythm. A CARTO (J&J MedTech) system was used to obtain the values during the procedure, and the open-source OpenEP package [38] was used to extract local activation maps post-procedure. EAM data was analyzed to identify each patient's earliest LA activation sites (first 5 ms) and total activation time.

Total activation times were used to personalize organ-scale CV values through an iterative inversion procedure. In this procedure, we fixed the anisotropy ratio for longitudinal CV ($CV_L$) versus transverse CV ($CV_T$) at $1 : 1/\sqrt{5}$ [14,15], limiting the optimization to a single varying parameter ($CV_L$). The optimization was performed by running EP simulations with our computational model, comparing the simulated to the recorded total activation time, and then adjusting the model's CV values accordingly. This process was repeated until the difference between model-predicted and recorded activation times was less than 1 ms.

Activation times, electrical stimulus locations, and resulting simulated activation are displayed in Fig 2, while resulting CV values are listed in Table 1. Reported values were representative for the whole tissue, without any adjustments for fibrotic regions. Additional details are provided as supplementary material, including detailed descriptions of the procedure (Sect A in S1 Text) and extended figures showing activation and electrical stimulus locations from multiple angles (Figs A-C in S1 Text).

## 2.5 Multi-scale, multi-physics LA modeling framework

To predict changes in atrial function under different conditions, we employed a multi-scale, multi-physics LA modeling framework. Electrical activation and mechanical contraction were modeled using a weakly coupled 3D electromechanical framework [21,39,40], with deformation strongly coupled to a 0D circulatory model [41]. Pulmonary veins were assumed to share the properties as non-fibrotic myocardium, exhibiting relative shortening within literature ranges [42].

Electrical activation was first modeled through a multiscale EP model, coupling cell- and tissue-scale dynamics. The cell-scale model was based on the human atrial action potential by Courtemanche et al. [43] with modifications according to Bayer et al. [44]. We simulated tissue-level electrical propagation using a reaction-eikonal model [45] with diffusion. Personalized CV values derived from EAM data (as described in Sect 2.4) were prescribed, for which baseline values are listed in Table 1. Pulmonary vein caps and the mitral valve were modeled as an in-excitable and non-conductive material.

The intracellular calcium distributions predicted by the EP framework was used as input to the biomechanical model. Active tension generation caused by intracellular calcium were computed using the cell-level contraction model by Land et al. [46,47]. Maximum active tension ($T_a$) was scaled with 50 kPa at baseline, calibrated to give approximately a 30% active LA emptying fraction (LAEF) [17,48–50]. Passive tension was modeled using a reduced Holzapfel-Ogden formulation with dispersion, with default material parameters $a = 2.92$ kPa, $b = 5.6$, $a_f = 11.84$ kPa, $b_f = 17.95$, and $\delta_f = 0.09$ [21,40]. Pulmonary caps and the mitral valve were modeled as a passive, stiff Demiray material [51], with material parameters $a = 10\,000$ kPa, $b = 5.6$. Both materials were modeled as nearly incompressible, with bulk modulus $\kappa = 650$ kPa [21].

To account for the impact of blood flow pressure and load from ventricular contraction, the 3D mechanical model of the LA was strongly coupled [41] with the 0D circulatory model CircAdapt [52,53]. Coupling between the 3D and 0D models was achieved by simulating 0D blood flow through the pulmonary veins into the LA and through the mitral valve

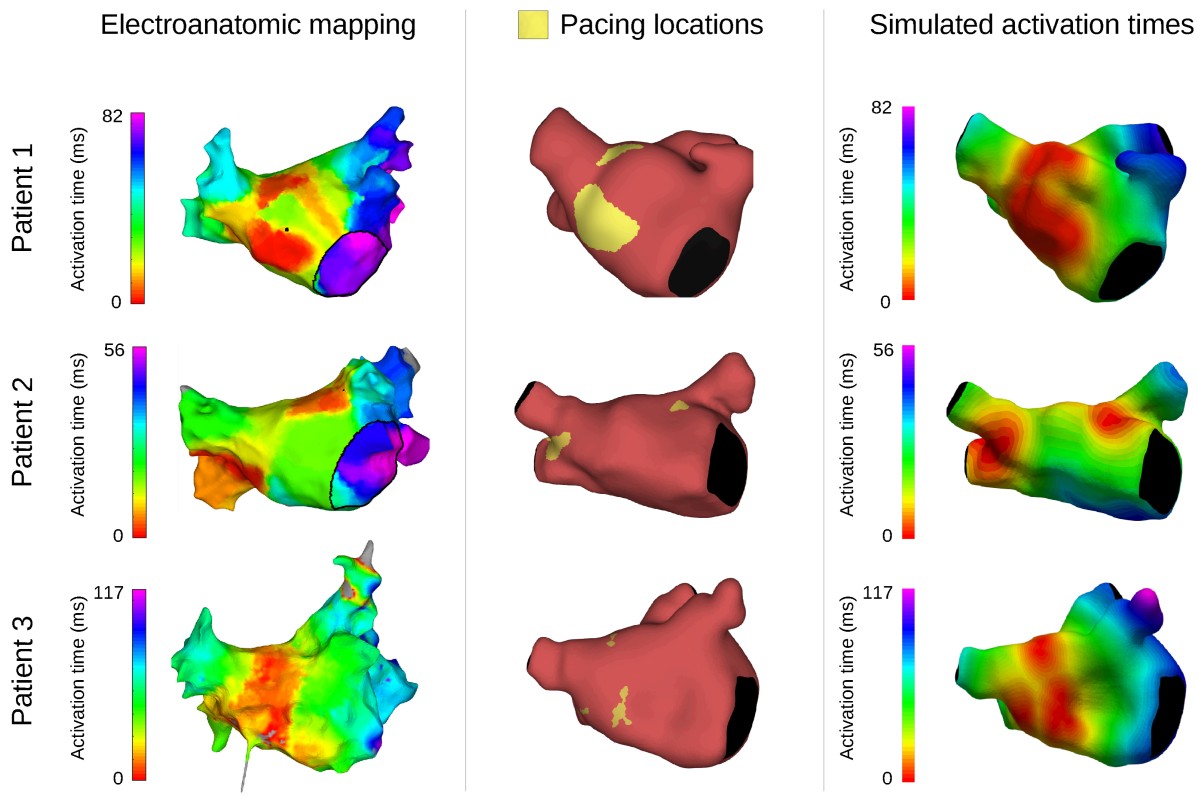

**Fig 2**. **Electroanatomical mapping (EAM) activation time, electrical stimulation locations, and model activation times.** Activation times estimated from EAM data (left), electrical stimulus locations corresponding to areas with the earliest activation times (first 5 ms, middle), and resulting activation times predicted by our EP model (right). All activation times are displayed relative to first LA activation.

into the left ventricle, thereby modulating the pressure applied to the endocardial surface of the 3D LA model. Active tension generated in the 0D left ventricle was used to model atrioventricular plane displacement by applying a traction boundary condition on the mitral valve annulus of the 3D model. Additional mechanical constraints not captured in the 0D CircAdapt framework were imposed using Robin-type boundary conditions: spatially varying, normal spring boundary conditions were applied to the epicardial surface [54], while omnidirectional spring boundary conditions were imposed at the pulmonary vein inlets.

## 2.6 Simulation details

Simulations were performed using the software *Carpentry* (Numericor GmbH, Graz, Austria) [39], which is built on extensions of the open-source EP platform *openCARP* [55]. All simulations were carried out on Hyak, the University of Washington's high-performance computing cluster (on nodes with a Intel(R) Xeon(R) Gold 6230 CPU processor), in parallel using 40 processes. The steps taken to perform a full integrated simulation workflow are described below. All scripts used for the CARPentry simulations are publicly available [37].

We initially performed an unloading-reloading procedure to establish a physiologically accurate end-diastolic pressure configuration. This process ensured that the reinflated, pressurized geometry point-wise matched the initial mesh generated from image data. Specifically, we applied a backward displacement algorithm [56] to determine the unloaded reference configuration using a prescribed diastolic pressure of 10 mmHg [17,40], and reloaded this reference geometry with the same diastolic pressure.

To ensure physiologically reasonable initial states, we next performed electromechanical simulations at the cellular level using *openCARP*'s single-cell tool *bench*. These simulations were run for 50 cardiac cycles with a fine temporal resolution of 0.025 ms. Leveraging the substantially lower computational cost of cell-level simulations compared to full organ-scale models, this approach allowed efficient approximation of steady-state conditions. The resulting cellular states were then used to initialize the full organ-scale simulations.

Organ-scale simulations were then performed for ten cardiac cycles with the full 3D framework. The ten cycles were simulated to reach a convergent state (convergence plots for baseline simulations displayed in Fig 3; in the 10th cycle there was less than 2% difference for both pressure and volume), balanced with a reasonable running time (16–20 hours per simulation).

The simulated end-diastolic volumes were slightly higher than the corresponding clinical measurements (Table 1; Fig 3, middle row). This was expected given that the volume computations in the simulations were performed on a closed atrial cavity that included the pulmonary vein remnants, see Fig 1. Specifically, the simulations yielded end-diastolic LA volumes of 124.2 mL, 130.9 mL, and 63.4 mL, compared to clinically measured volumes of 110.0 mL, 112.5 mL, and 60.0 mL. This systematic offset was consistent with the modeling approach and demonstrated an overall agreement between simulated and clinical data.

For each cycle, activation was initiated by applying an electrical stimulus in earliest activation regions, determined by EAM data (as described in Sect 2.4). The remaining myocardial tissue was then activated through the propagation of the electrical signal, leading to atrial contraction as simulated by our modeling pipeline. We imposed a 1 Hz frequency to mimic sinus rhythm. Time steps were discretized at 0.025 ms for the EP model and 0.5 ms for the coupled biomechanical-0D circulatory model. The PV loops predicted by the final cycle simulated were used for our subsequent analysis.

## 2.7 Parameter changes in fibrotic regions

In our investigation of consequences of various fibrosis-related properties, we focused on five EP parameters (CV values and ion channel currents) and four mechanical parameters (contractile and passive properties), summarized in Table 2. Each parameter was re-scaled in the fibrotic regions by the corresponding factor (second column). Baseline or fibrotic parameter values were assigned in various combinations following the sensitivity analyses described in Sect 2.8. An overview of all parameter absolute values across all combinations is publicly available [37].

For CV values, we independently explored reductions in the longitudinal and transverse directions. $CV_L$ was scaled a factor of 0.657, based on values reported by Macheret et al. [16]. For $CV_T$, we used a scaling factor of 0.520 (relative to baseline $CV_T$ values), corresponding to a fibrotic anisotropy ratio of $1 : 1/\sqrt{8}$. This imposed higher anisotropy compared to the non-fibrotic ratio $1 : 1/\sqrt{5}$, consistent with previous studies [14,15].

Changes in ionic currents were imposed by scaling the inward rectifier potassium ($I_{K1}$), L-type calcium ($I_{CaL}$), and fast sodium ($I_{Na}$) currents by factors of 0.5, 0.5, and 0.6, respectively [14,15,57–59]. These changes in ionic current levels

**Table 2**. Fibrotic remodeling parameter changes.

| Parameter | Scaling factor(s) | Notes | References |
|---|---|---|---|
| CV longitudinal direction ($CV_L$) | 0.657 | | [16] |
| CV transverse direction ($CV_T$) | 0.520 | Anisotropy ratio $1 : 1/\sqrt{8}$ | [14,15] |
| Inward rectifier potassium current ($I_{K1}$) | 0.5 | Also alters CV values | [14,15,57–59] |
| L-type calcium current ($I_{CaL}$) | 0.5 | Also alters CV values | [14,15,57–59] |
| Fast sodium current ($I_{Na}$) | 0.6 | Also alters CV values | [14,15,57–59] |
| Myofilament binding rate ($\mu$) | 0.5 | | [60,61] |
| Active tension scaling factor ($T_a$) | 0.5 | | [21,62] |
| Increased longitudinal stiffness ($ST_L$) | $a \times 0.963$, $a_f \times 2.293$ | $2 \times$ load upon 5% stretch longitudinal direction | [7,62] |
| Increased transverse stiffness ($ST_T$) | $a \times 2.157$, $a_f \times 0.675$ | $2 \times$ load upon 5% stretch transverse direction | [7,62] |
| $\Rightarrow ST_L$ and $ST_T$ in combination | $a \times 2.009$, $a_f \times 2.000$ | | |

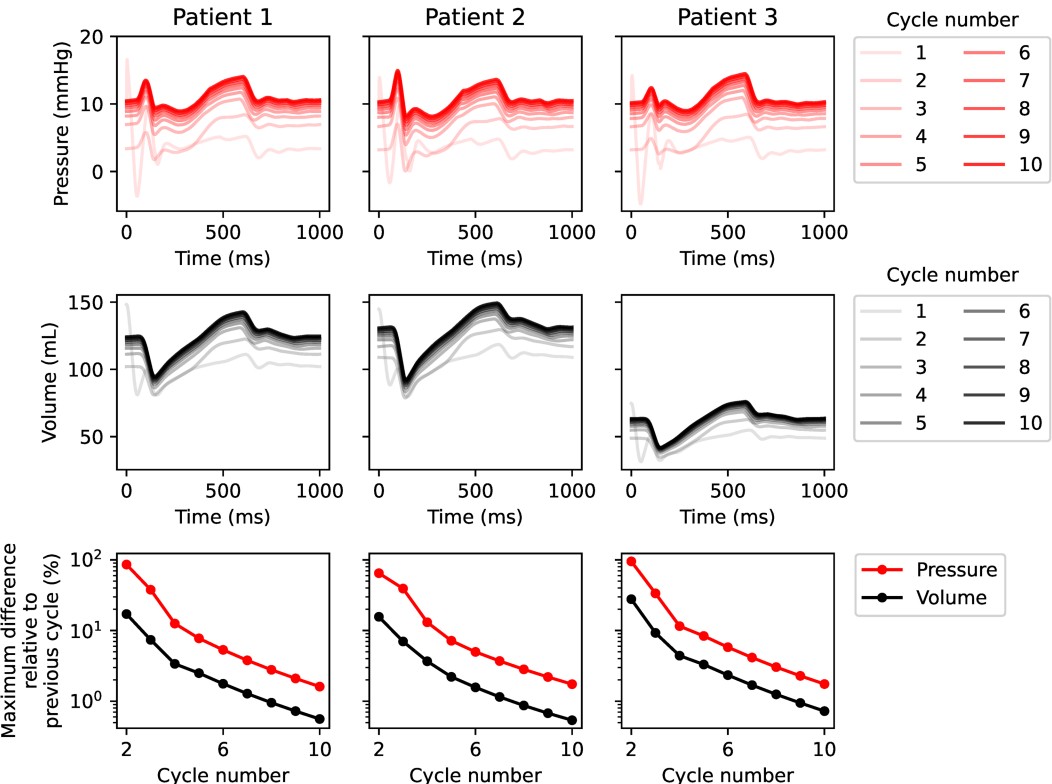

**Fig 3. Convergence plots for baseline simulations.** LA pressure (top), volume (middle) over time for ten cardiac cycles, and maximum differences in respective values between subsequent cardiac cycles, normalized by the maximum value of the last cycle (bottom). Values shown are shown for Patient 1–3, at baseline.

are known to alter CV values locally [63,64]. In traditional tissue- and organ-scale EP models, these alterations would be captured indirectly through changes in membrane potential. However, the reaction-eikonal formulation we used in our study does not inherently account for the direct effects of ionic current changes. CV values are here prescribed and not emergent properties of the underlying cell- and tissue-scale EP dynamics. To prescribe accurate CV values, we ran supplement simulations using OpenCarp's *tuneCV* functionality [65]. These simulations were performed considering a simplified 1D rod to predict appropriate CV scaling factors for all combinations of ionic scaling factors. These are described in Sect B of S1 Text, and the corresponding scripts are published [37]. The resulting CV values for all possible parameter combinations, including in combination with reduced $CV_L$ and $CV_T$, are listed in Table A in S1 Text.

For contractile properties, we considered the active tension scaling factor ($T_a$) and the myofilament binding rate ($\mu$). Modifications to $T_a$ affect the magnitude of the active transient. We applied a fibrotic scaling factor of 50% for $T_a$ [21,62] (relative to the baseline value of 50 kPa). The myofilament binding rate depends on the ratio of $\alpha$ and $\beta$ myosin isoforms. This ratio shifts to a higher proportion of $\beta$ myosins in AF [60,61], resulting in a slower contraction rate. We here imposed a fibrotic scaling factor of 50% (relative to the baseline value $\mu = 9$ [47]).

To account for increased myocardial stiffness, we altered passive material parameters imposing changes in longitudinal ($ST_L$) and transverse ($ST_T$) stiffness. Specifically, we altered parameters $a$, which affects isotropic stiffness, and $a_f$, which alters additional fiber direction stiffness. To consider stiffness independently for each direction, we performed virtual stretch experiments. Through these experiments we estimated values of $a$ and $a_f$ needed to achieve a two-fold change

in the load (i.e., force per area) in one direction while keeping the load in the other direction unchanged. In these experiments, we stretched a tissue block (a unit cube) by 5% in either direction while tracking the corresponding load values, following the setup described in a previous publication [66]. Through an iterative procedure, we then stretched the tissue block, compared new load values with previous values, then altered the values of $a$ and $a_f$. We repeated the process until the load value doubled upon stretching in one direction while remaining unchanged in the other direction (within 1% precision). For the combined effect, having both $ST_L$ and $ST_T$ set to fibrotic levels, we found values of $a$ and $a_f$ such that load values doubled when stretched 5% in either direction. Resulting $a$ and $a_f$ values are listed in Table 2. Unloading-reloading procedures were performed for each geometry at both levels of fibrosis and for all four stiffness configurations (without any changes, with increased $ST_L$, with increased $ST_T$, and both combined). The code for material parameter estimation is publicly available on Zenodo [67].

## 2.8 Sensitivity analysis – experimental setup

We investigated the impact of the nine identified fibrosis-associated parameters using two approaches. First, we used an OFAT analysis to analyze each parameter's isolated effect. Next, we used a FFD analysis to evaluate combined effects. We employed a $2^{9-5}$ design [68] as displayed in Table 3, resulting in 32 distinct parameter combinations. In this setup, B denotes the baseline factor level, while F represents the fibrosis-associated factor level, incorporating the relevant scaling factors (listed in Table 2). The code for performing the post-simulation sensitivity analysis is publicly available on Zenodo [69], and the pressure and volume data from all simulations are available on Dryad [37].

## 2.9 Metrics reported

For all simulations, we reported five PV-loop-based metrics capturing different aspects of atrial function. These included A-loop area [70–72], booster function, reservoir function, conduit function [73,74], and upstroke pressure difference during contraction. The metrics are defined in the equations given below. Here, $LAV_{preA}$ and $LAP_{preA}$ refer to the LA volume and pressure at the time point for initial electrical stimulus (LA pre-systolic/LA end-diastolic volume), $LAV_{min}$ and $LAV_{max}$ refer to the minimal (LA end-systole) and maximal volumes, and $LAP_{maxA}$ refers to the maximal pressure during the A-loop (i.e., maximum value of booster phase pressure). We performed statistical analysis on values normalized by the corresponding baseline values for the same patient.

The five metrics considered were:

1. A-loop area (work performed during the active contraction), calculated by Gauss's area formula:

$$A = \frac{1}{2}\sum_{i=1}^{n} p_i(v_{i+1} - v_{i-1}) = \frac{1}{2}\left(\sum_{i=1}^{n} p_i v_{i+1} - \sum_{i=1}^{n} p_{i-1} v_i\right) \tag{1}$$

**Table 3**. FFD setup; B = baseline value; F = fibrotic value.

| | Combination number | | | | | | | | | | | | | | | | | | | | | | | | | | | | | | | |
|---|---|---|---|---|---|---|---|---|---|---|---|---|---|---|---|---|---|---|---|---|---|---|---|---|---|---|---|---|---|---|---|---|
| | 1 | 2 | 3 | 4 | 5 | 6 | 7 | 8 | 9 | 10 | 11 | 12 | 13 | 14 | 15 | 16 | 17 | 18 | 19 | 20 | 21 | 22 | 23 | 24 | 25 | 26 | 27 | 28 | 29 | 30 | 31 | 32 |
| $CV_L$ | B | F | B | F | B | F | B | F | B | F | B | F | B | F | B | F | B | F | B | F | B | F | B | F | B | F | B | F | B | F | B | F |
| $CV_T$ | B | B | F | F | B | B | F | F | B | B | F | F | B | B | F | F | B | B | F | F | B | B | F | F | B | B | F | F | B | B | F | F |
| $I_{K1}$ | B | B | B | B | F | F | F | F | B | B | B | B | F | F | F | F | B | B | B | B | F | F | F | F | B | B | B | B | F | F | F | F |
| $I_{CaL}$ | B | B | B | B | B | B | B | B | F | F | F | F | F | F | F | F | B | B | B | B | B | B | B | B | F | F | F | F | F | F | F | F |
| $I_{Na}$ | B | B | B | B | B | B | B | B | B | B | B | B | B | B | B | B | F | F | F | F | F | F | F | F | F | F | F | F | F | F | F | F |
| $\mu$ | F | F | B | B | B | B | F | F | B | B | F | F | F | F | B | B | B | B | F | F | F | F | B | B | F | F | B | B | B | B | F | F |
| $T_a$ | F | B | F | B | B | F | B | F | B | F | B | F | B | F | B | F | B | B | F | B | F | B | F | B | F | B | F | B | F | B | F | B |
| $ST_L$ | F | B | B | F | F | B | B | F | B | F | F | B | B | F | F | B | B | F | F | B | B | F | F | B | B | F | F | B | B | F | B | F |
| $ST_T$ | F | B | B | F | B | F | F | B | F | B | B | B | F | B | F | F | B | B | F | F | B | F | B | B | F | B | F | F | B | F | B | F |

where $p$ and $v$ correspond to A-loop volume and pressure values.

2. Booster function (LA pumping function, active contraction volume change):

$$\frac{LAV_{preA} - LAV_{min}}{LAV_{preA}} \tag{2}$$

3. Reservoir function (elastic ability to stretch and recoil, passive stretching):

$$\frac{LAV_{max} - LAV_{min}}{LAV_{min}} \tag{3}$$

4. Conduit function (passive transfer of blood to the LV):

$$\frac{LAV_{max} - LAV_{preA}}{LAV_{max}} \tag{4}$$

5. Upstroke pressure difference (LA pumping function, active contraction pressure change):

$$\frac{LAP_{maxA} - LAP_{preA}}{LAP_{preA}} \tag{5}$$

## 2.10 Spatiotemporal analysis

We examined spatiotemporal distributions of electromechanical simulations to better understand the effects of the parameters identified as most influential. Specifically, we compared the baseline simulation, those in which $I_{CaL}$ and $I_{K1}$ were impaired (from the OFAT analysis), and FFD Combination 32, in which all parameters were set to fibrotic levels. For conciseness, we refer to this combination as "Fully fibrotic" throughout the paper. We compared spatiotemporal distributions of membrane voltage, intracellular calcium, active tension, and fiber direction strain. This analysis was limited to the original fibrosis burden, with Patient 1 as a representative example.

## 2.11 FFD analysis

To analyze the FFD main effect (isolated impact of a single parameter, but in combination with other parameter changes), we compared the baseline (B) and fibrotic (F) groups. For a given parameter, the B group included all combinations in which that parameter was set to baseline value, while the F group includes combinations where it was set to the fibrotic value (as defined in Table 2). Other parameters were set to either baseline or fibrotic levels according to the design. Statistical comparisons between the two groups were performed using Student's t-test (with group-wise distributions found to be normal, as assessed by an Anderson-Darling test). The analysis was conducted and annotated using the software Statannotations [75].

To characterize FFD interaction effects (confounded effects with other parameters), the subdivision was extended to pairwise combinations. This resulted in four groups for each parameter pair $(x_i, x_j)$ – all combinations of baseline/fibrotic levels ($B_iB_j$, $F_iB_j$, $B_iF_j$, and $F_iF_j$). We defined an interaction coefficient as a cross-product between these:

$$I_{i,j} = \arcsin\left(\frac{[F_iB_j, F_iF_j] \times [B_iB_j, B_iF_j]}{\|[F_iB_j, F_iF_j]\| \; \|[B_iB_j, B_iF_j]\|}\right). \tag{6}$$

The coefficient relates the angle between two lines in an interaction plot. It is symmetric, meaning $I_{i,j} = I_{j,i}$. Positive values indicate a *synergistic effect*, where the impact of having both variables at fibrotic levels leads to a greater reduction in

atrial function than the sum of the effects of individual factors. Negative values indicate *mitigating effect*, where the impact of having both variables at fibrotic levels leads to a smaller reduction in atrial function compared to the sum of the effects of individual factors.

# 3 Results

In this section, we present findings from individual patient-specific simulations, using Patient 1 as a representative example, and statistical analysis of aggregated data. We begin by presenting an overview of the simulation pipeline, PV loops with derived metrics for all three patients, followed by OFAT analysis results. We next present results from the spatiotemporal analysis focusing on parameter changes emerging as most important from the OFAT analysis, followed by FFD analysis results. Finally, we examine the impact of increased fibrosis. Unless otherwise noted, the results are representative of all three patients and reflect the original fibrosis burden.

## 3.1 Electromechanical simulations and PV loop-based metrics

Our modeling pipeline is demonstrated in Fig 4, for Patient 1 at baseline (no fibrotic changes). The model-predicted pressure and volume transients were extracted, resulting in characteristic LA PV loops. The A-loop was larger than the V-loop in area, indicating higher changes in volume and pressure. Prior to atrial contraction (when we were close to an equilibrium between atrial and ventricular pressure), we observed small oscillations in both volume and pressure, also are visible in the lower part of the V-loop.

Baseline PV loops for all three patients are displayed in Fig 5, with derived metric values annotated. Patient 1 had the smallest booster and reservoir function. Patient 2 had the largest A-loop area and upstroke pressure difference, and the lowest conduit function (although only marginally lower than Patient 1). Patient 3 had the smallest A-loop area and upstroke pressure difference, but the highest booster, conduit, and reservoir function. No patient consistently showed higher metric values than the others.

Based on volumetric measurements at different stages, we also computed model-predicted LAEF for comparison with clinical values (reported in Table 1). Total LAEFs at baseline were 33.70% for Patient 1, 38.13% for Patient 2, and 37.52%; consistently lower than corresponding clinical estimates. Simulation-predicted active LAEFs were 23.99%, 29.37%, and 33.97%, while passive LAEFs were 12.78%, 12.16%, and 16.61% for Patients 1–3.

## 3.2 OFAT analysis predicted fibrosis-associated decreases in $I_{CaL}$ and $I_{K1}$ as main determinants of LA function

Results of our OFAT analysis are presented in Fig 6, highlighting the isolated effect of each parameter change. Fig 6A shows the impact on PV loops using Patient 1 as a representative example. Impaired $I_{K1}$ enlarged the A-loop, while impaired $I_{CaL}$ (substantially) and reduced $T_a$ (marginally) decreased it. Increased stiffness (i.e., higher $ST_F$ and $ST_T$) shifted the PV loop, increasing volume while preserving pressure and loop shape. Modifying $CV_L$, $CV_T$, $I_{Na}$, or $\mu$ did not noticeably alter PV loop size or shape.

Relative changes in PV-loop-derived metrics are plotted in Fig 6B. There were consistent trends across all three patients, with comparable magnitude changes. The largest changes were observed for Patient 2 (who had the highest fibrosis burden). A-loop area was the most sensitive metric (at maximum leading to a 74% reduction for impaired $I_{CaL}$ for Patient 2).

Fig 6C displays output metrics and fibrosis-associated input parameters averaged across all three patients. Reducing $I_{CaL}$ substantially decreased atrial function (with the largest decrease in A-loop area, 64%), while reducing $I_{K1}$ increased function (with the largest increase in A-loop area, 20%). Reducing $T_a$ also had an effect (17% decrease in A-loop area). The impact of the other factors was otherwise modest. Notably, reduced CV and increased stiffness did not alter any metric by more than 5%.

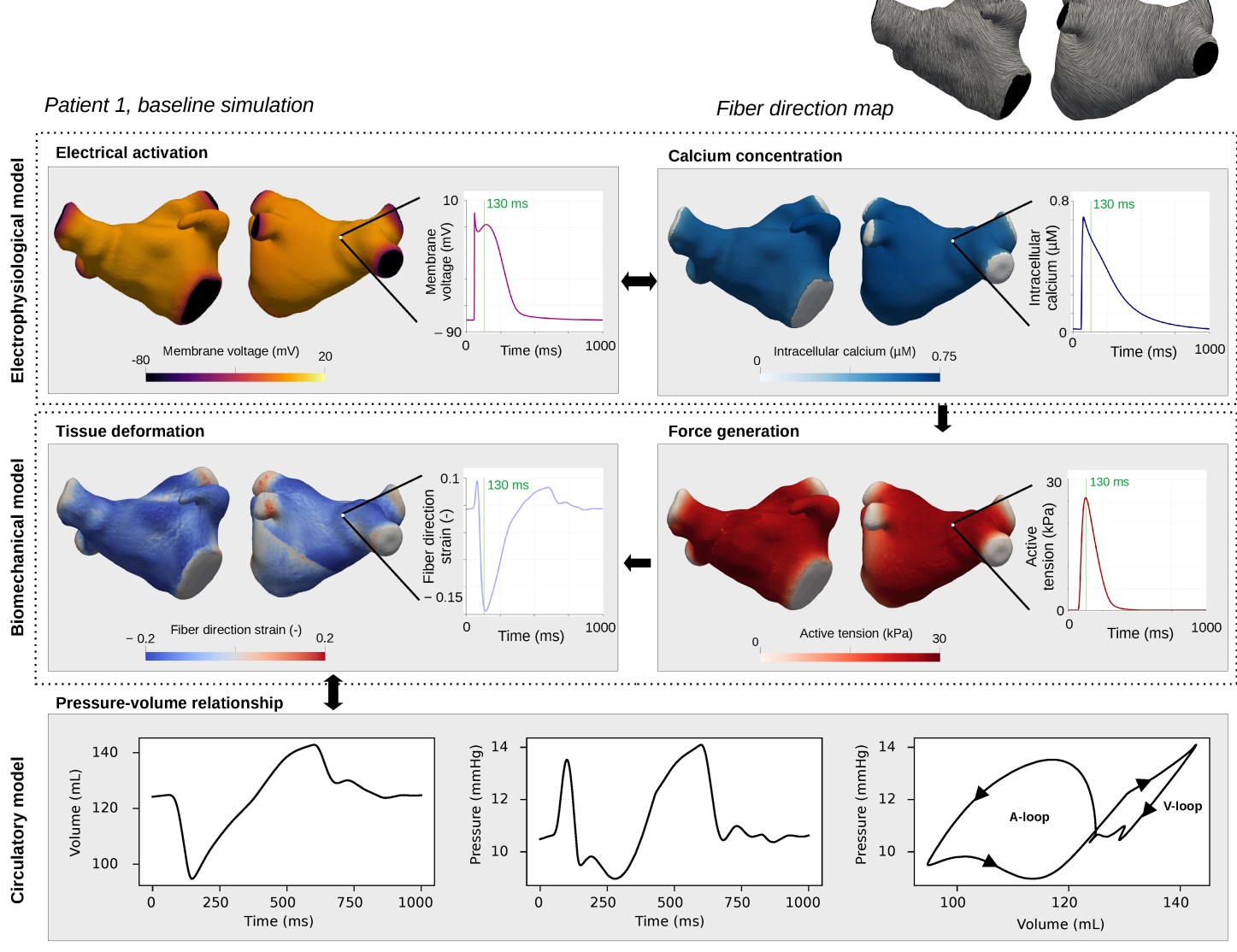

**Fig 4**. **Model pipeline: EP, biomechanical, and circulatory models.** Electrical propagation, including the release of intracellular calcium, was modeled with the EP model. The intracellular calcium resulted in active tension generation in our biomechanical model, leading to tissue deformation. Tissue deformation was strongly coupled to the 0D circulatory model, from which we extracted LA volume and pressure transients. 3D maps (left) show the spatial distribution of model outputs at the 130 ms time point, and transient plots (right) display values over time at a representative node. Output data are shown for Patient 1, baseline simulation (no fibrotic changes). Movies displaying spatiotemporal distributions for all three patients are included as supplementary material (S1 Movie, S2 Movie, and S3 Movie).

### 3.3 Impact of impaired $I_{CaL}$ and $I_{K1}$ was related to changes in intracellular calcium transient amplitude

Motivated by the results from the OFAT analysis, we next compared key spatiotemporal distributions across baseline, impaired $I_{CaL}$, impaired $I_{K1}$, and fully fibrotic simulations. Spatiotemporal distributions for all three patients under these conditions are also included as supplementary material (S1 Movie, S2 Movie, and S3 Movie).

Fig 7 shows membrane potential, intracellular calcium, and active tension for each condition, using Patient 1 as a representative example. In the simulation with impaired $I_{CaL}$, membrane potential was lower in fibrotic areas, while

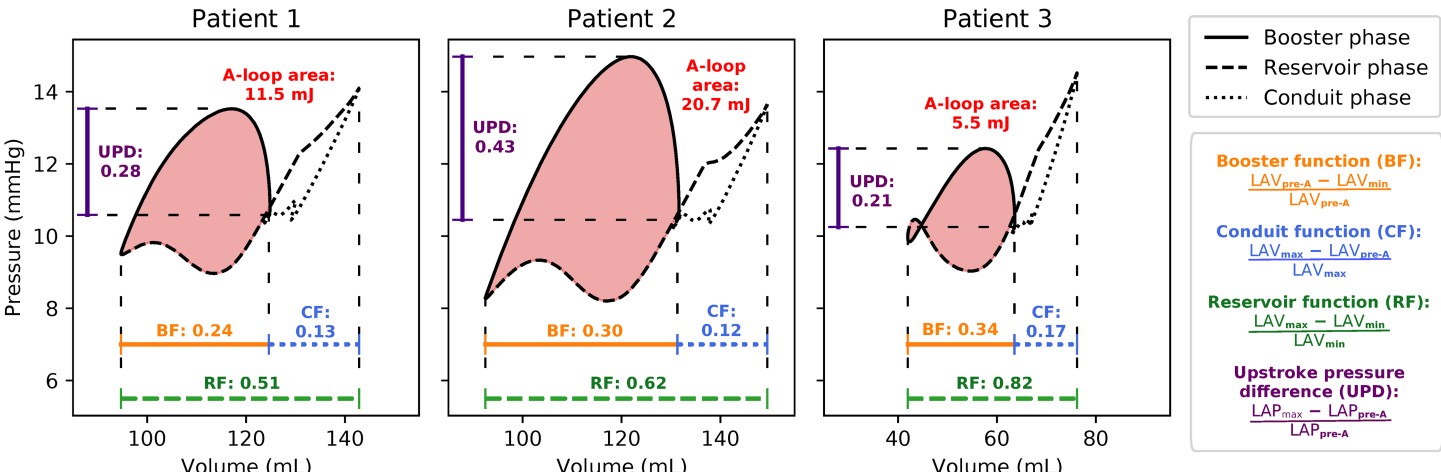

**Fig 5. PV loops and derived metrics.** PV loops from baseline simulations for all three patients, with derived metrics annotated: A-loop area (i.e., stroke work in mJ), booster function (BF), conduit function (CF), reservoir function (RF), and upstroke pressure difference (UPD).

intracellular calcium was intermediate, and active tension was close to zero. A dispersive effect was also observed for calcium and active tension, lower in non-fibrotic areas than in the baseline simulation (compare, e.g., the lower left region). In contrast, impaired $I_{K1}$ abolished the differences between fibrotic and non-fibrotic regions producing a slight increase in intracellular calcium and a marked rise in active tension relative to baseline (see also transients in Fig 8). The fully fibrotic simulation exhibited pronounced heterogeneity in the intracellular calcium distribution, with near-zero values centrally and intermediate closer to non-fibrotic areas. Active tension was close to zero in all fibrotic regions, but higher in non-fibrotic areas compared to the impaired $I_{CaL}$ simulation.

Fig 8 shows membrane potential, intracellular calcium, and active tension transients for three representative locations in Patient 1's geometry. The fibrotic site and its immediate vicinity (Points A and B) exhibited an elevated resting membrane potential (approximately 5 mV increase) and a prolonged action potential duration (approximately 75 ms longer) in the impaired $I_{K1}$ and fully fibrotic simulations. No differences were observed at the distant site (Point C) for the membrane potential. Intracellular calcium transient amplitude was reduced for the impaired $I_{CaL}$ simulation and slightly increased for the impaired $I_{K1}$ simulation across all points, with gradual effect from Point A to Point B to Point C. In the fully fibrotic simulation, intracellular calcium was close to zero in Point A (see also corresponding spatial plots in Fig 7, bottom middle) while being reduced in Point B (14% decrease in amplitude, relative to baseline) and Point C (7% decrease). These calcium variations were magnified in active tension, with attenuated responses at lower calcium levels and enhanced responses at higher calcium levels. For the fully fibrotic simulation, active tension was zero in Point A, while reduced in Point B (44% decrease, relative to baseline) and Point C (19% decrease).

In Fig 9A, we show LA deformation comparing baseline to the other configurations. Deformation is displayed at the time of minimum volume (i.e., LA systole, see Fig 9B, top subplot). In the impaired $I_{CaL}$ simulation (top row), the LA remained more dilated, indicating reduced contraction. Conversely, for impaired $I_{K1}$ (middle row), the deformed geometry was slightly more contracted than the baseline simulation. The fully fibrotic simulation (bottom row) also resulted in a less contracted geometry.

Fig 9B displays corresponding differences in volume and pressure over time. The systolic volumes were similar between impaired $I_{CaL}$ and fully fibrotic simulations, while the fully fibrotic simulation exhibited a larger maximum volume. Pressure displayed highest differences during contraction, with the impaired $I_{CaL}$ and fully fibrotic simulations having less variation than the baseline and impaired $I_{K1}$ ones. The PV loop in the fully fibrotic simulation was shifted in

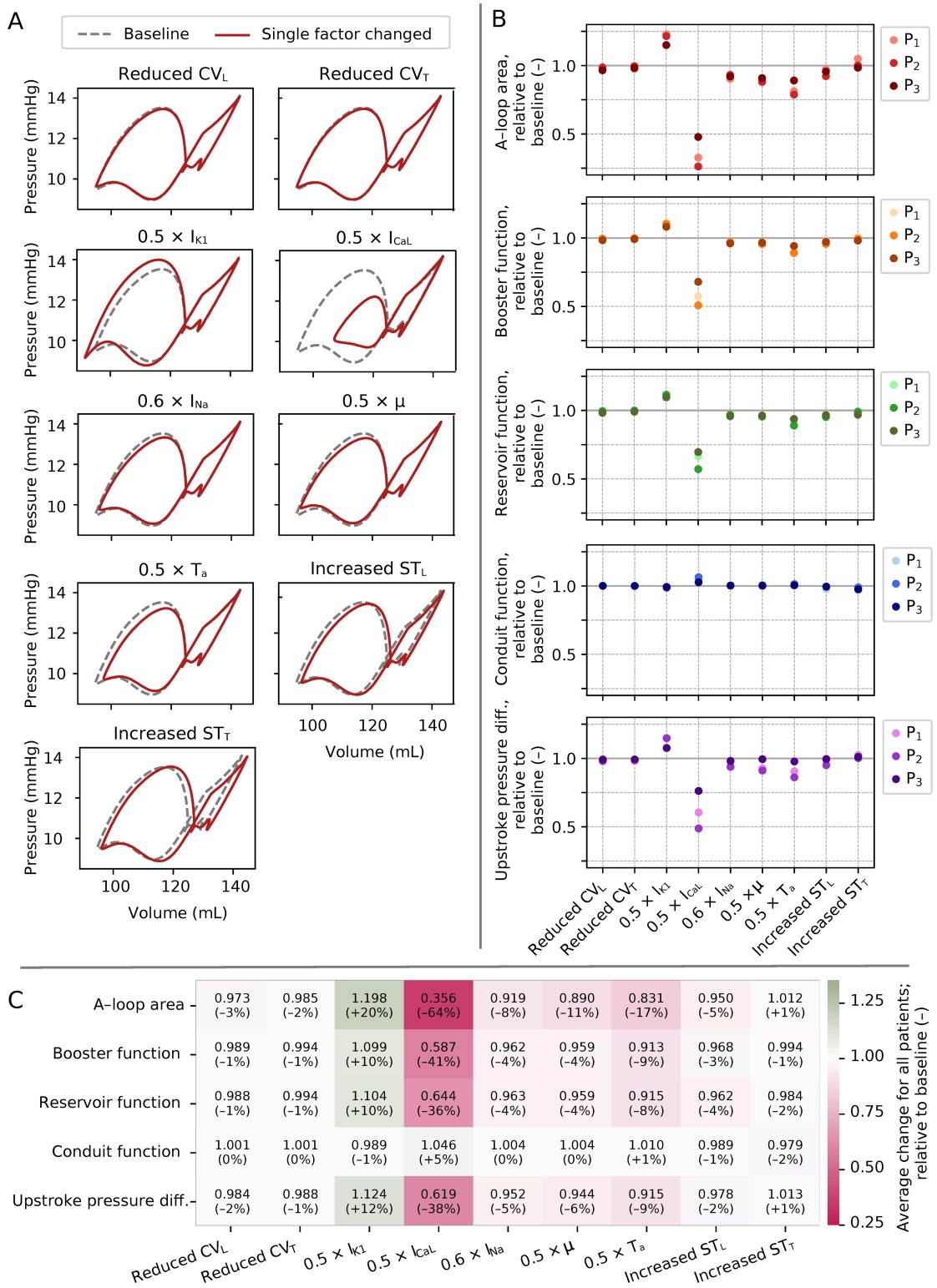

**Fig 6. Results from OFAT analysis, original fibrosis burden.** (A) PV loops with one parameter changed (red) compared to baseline (gray, dotted), for Patient 1. (B) Impact of each parameter on A-loop area, booster function, conduit function, reservoir function, and upstroke pressure difference for Patients 1–3 ($P_1$–$P_3$). Values reported relative to baseline. (C) Average change (across all three patients) in each metric for each parameter, with implied percent-wise changes in parentheses.

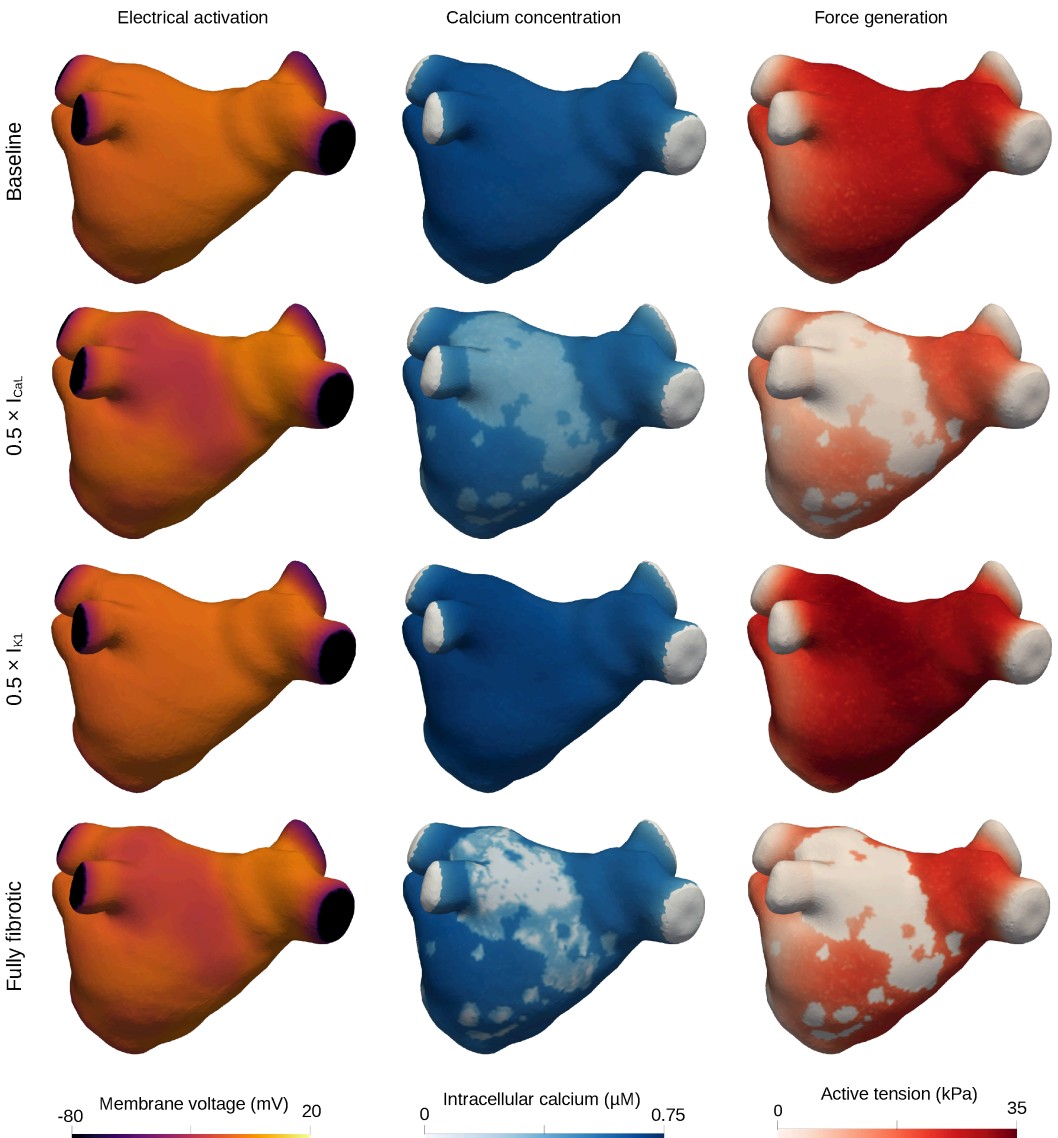

**Fig 7**. **Spatial distributions of membrane potential, intracellular calcium, and active tension; baseline, impaired I_CaL, impaired I_K1, and fully fibrotic simulations.** Spatial distributions of membrane potential (left), intracellular calcium (middle), and active tension (right) at time step 130 ms for baseline (top), impaired I_CaL (second row), impaired I_K1 (third row), and fully fibrotic (bottom row) simulations for Patient 1. See also corresponding point-wise plots displayed in Fig 8 and movies attached as supplementary material (S1 Movie, S2 Movie, and S3 Movie).

volume compared to all the other cases, and the A-loop was smaller than baseline and impaired I_K1 but larger than in the I_CaL simulation.

### 3.4 FFD analysis predicted I_CaL, I_K1, longitudinal, and transverse stiffness as significant factors

To gain deeper understanding of the interactions between model parameters, we next performed a detailed FFD analysis. The PV loops obtained from simulations for all FFD combinations are shown in Fig 10, in which most combinations reduced the A-loop. Patient-averaged metric values for all combinations are included in Fig D in S1 Text, and further described in Sect C in S1 Text. For Combination 32 (fully fibrotic), the relative change in A-loop area was 0.474

**Fig 8.** **Membrane potential, intracellular calcium, and active tension in selected points; baseline, impaired $I_{CaL}$, impaired $I_{K1}$, and fully fibrotic simulations.** Membrane voltage, intracellular calcium, and active tension transients for three representative points (Point A – inside the fibrotic area, Point B – close to the fibrotic area, and Point C – away from the fibrotic area), Patient 1. Note that active tension for Point A in the fully fibrotic simulation was zero everywhere (not visible in the plot). See also corresponding spatial plots displayed in Fig 8.

(i.e., −53%), in booster function 0.673 (−33%), in reservoir function 0.695 (−30%), in conduit function 1.0 (no change), and in upstroke pressure difference 0.721 (−28%).

Main effect plots for all patients and combinations are displayed in Fig 11A, with the relative change of each factor shown in Fig 11B. Reduced $I_{CaL}$ and $I_{K1}$ had a statistically significant impact on four of five metrics when comparing B and F groups, the impact in line with the OFAT analysis. The effect of impaired $I_{CaL}$ was slightly attenuated (e.g., 54% reduction in A-loop area vs. 64% reduction in the OFAT analysis), whereas the impact of $I_{K1}$ impairment was amplified (e.g., 27% increase in A-loop area vs. 20% increase in the OFAT analysis). Consistent with our OFAT analysis, changes in CV and stiffness had a modest impact (less than 2%).

Increasing stiffness had a statistically significant, albeit modest impact on conduit function, with decreases of 1% and 2% for increases in longitudinal and transverse stiffness, respectively. The impact was small in magnitude relative to the effects observed for the other metrics, with little variation (all standard deviations less than 0.03, and all observations were within the range [0.95,1.06]).

Interaction effects presented in Fig 12 were generally small. All interaction coefficients (Fig 12A) were less than 0.1 radians, and interaction lines nearly parallel (Fig 12B–12C). Most combinations were found to be mitigating (having negative interaction values, see again Fig 12A). For A-loop area, booster function, reservoir function, and upstroke

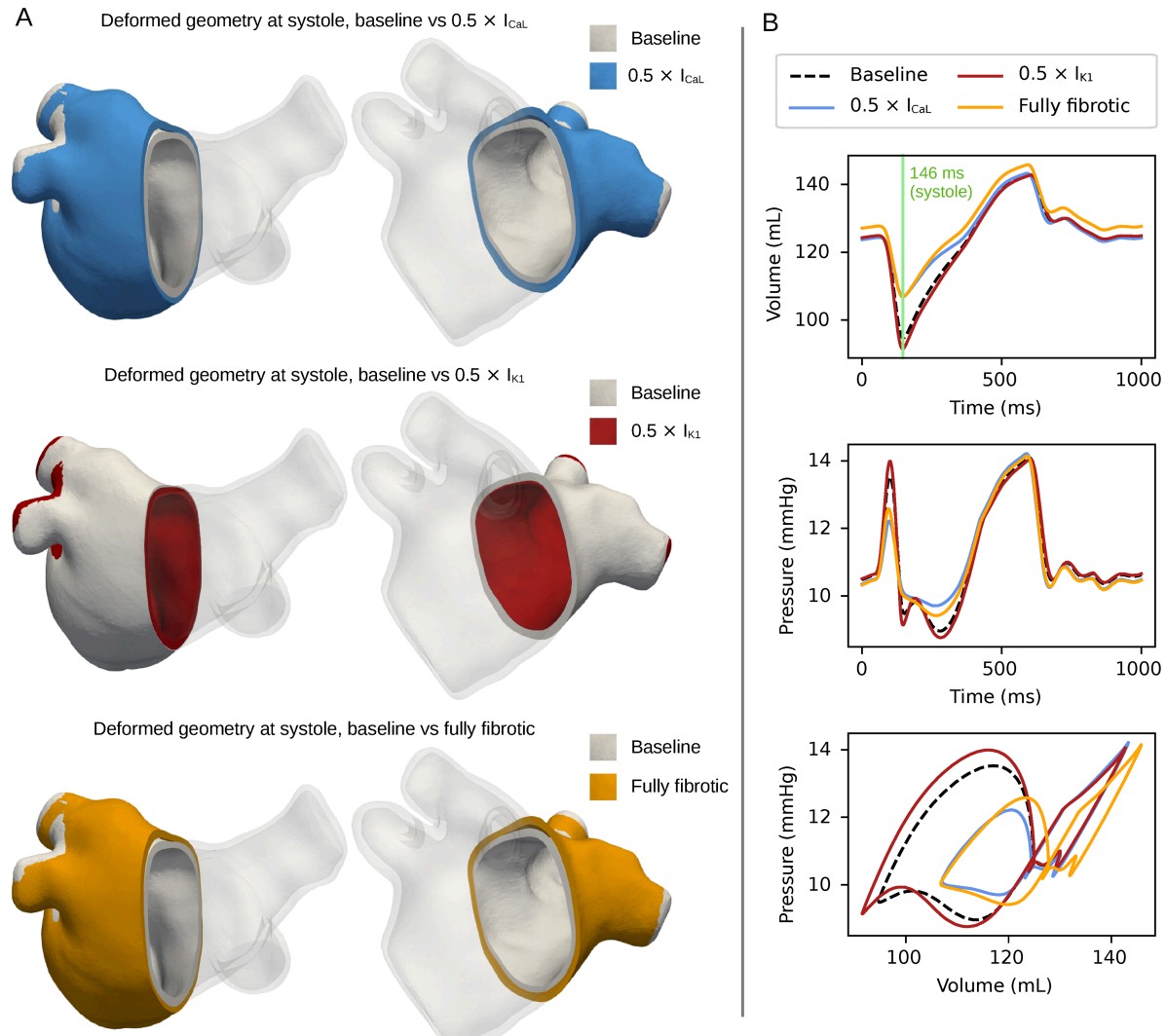

**Fig 9**. **Geometry deformation and changes in volume and pressure – comparison of baseline with impaired $I_{CaL}$, impaired $I_{K1}$, and fully fibrotic simulations.** (A) Deformed geometries for impaired $I_{CaL}$ vs. baseline (top), impaired $I_{K1}$ vs. baseline (middle), and fully fibrotic vs. baseline (bottom) simulations, Patient 1. Deformation is displayed from two angles, with partial transparency. (B) Volume over time, pressure over time, and PV loops for the same simulations.

pressure difference, the greatest mitigating effect consistently was for reduced $CV_T$ combined with increased $ST_T$. There was also a mitigating effect for $I_{CaL}$ combined with $I_{K1}$ – strongest for A-loop area, followed by booster function.

### 3.5 Increased fibrosis led to moderate further reduction in LA function

Results of our analysis with a 50% synthetically elevated fibrosis burden are presented in Fig 13. Increased fibrosis burden led to a further reduction across all metrics in the OFAT analysis (Fig 13A), however, the decrease was moderate. Factors identified as statistically significant for FFD main effect subject to original fibrosis burden remained significant with elevated fibrosis (Fig 13B). Considering average values (Fig 13C), the largest increase in absolute effect was observed

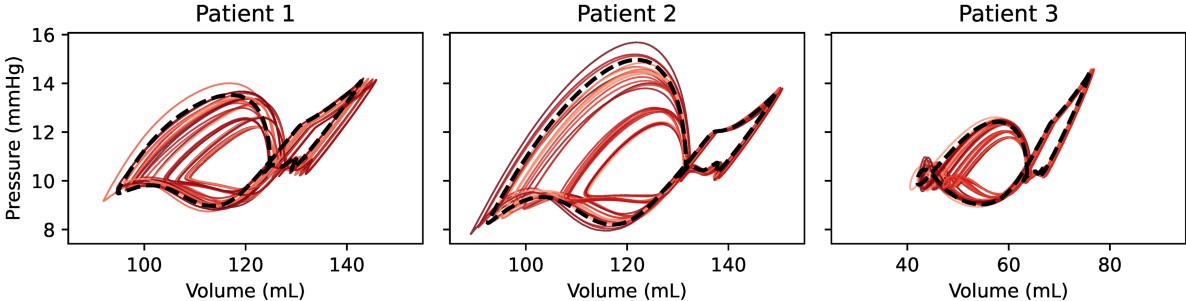

**Fig 10**. **PV loops for all FFD combinations, original fibrosis burden.** PV loops from all FFD combinations, for all three patients. Baseline PV loops are displayed with a bolder black dotted line, while the thinner red traces correspond to FFD combinations, as listed in Table 2.

for impaired $I_{CaL}$. For A-loop area, we observed a 74% decrease with elevated fibrosis (compared to 64% with the original fibrosis burden) in the OFAT analysis, and a 62% decrease (compared to 54%) in the FFD analysis. Patient-averaged metric values for all FFD combinations are included in Fig E in S1 Text, and described in Sect C in S1 Text. For Combination 32 (fully fibrotic), the relative change in A-loop area was 0.366 (i.e., −63%), booster function 0.584 (−42%), reservoir function 0.622 (−38%), conduit function 0.995 (+0.5%, more precisely +0.462 rounded down to 0 in the heatmap), and upstroke pressure difference 0.657 (−34%).

In Fig 14 we display how absolute values of all metrics vary between no fibrosis, original fibrosis, and increased fibrosis burden, considering the fully fibrotic simulations. The decrease in all but the conduit function was steeper going from no fibrosis to original fibrosis burden compared to going from original to elevated fibrosis burden. Patient-specific trends correlated well between A-loop area and upstroke pressure difference, and between booster function and reservoir function, while conduit function differed from all other metrics.

## 4 Discussion

### 4.1 Impact of fibrotic-associated parameter changes on the atrial function

In the present study, we combined a multi-scale, multiphysics modeling framework with patient-specific LA geometries and fibrosis maps to investigate the impact of fibrotic remodeling. We investigated the influence of nine electromechanical parameters altered in fibrosis, and assessed LA function using five volume- and pressure-based metrics. We found these metrics to be sensitive to variations in $I_{CaL}$ and $I_{K1}$, with opposite effects, while stiffness exerted a small but statistically significant impact on conduit function. Spatiotemporal analysis revealed that changes in intracellular calcium transient amplitude had pronounced effects on active tension impairment, reflecting the nonlinear relationship between these variables. In simulations with impaired $I_{K1}$, we observed an elevated membrane potential and an increase in intracellular calcium transient amplitude (Fig 8), leading to enhanced active tension and improved LA function. This effect mitigated the impairing impact of fibrosis from other factors.

The effect of $I_{CaL}$ on atrial function is consistent with its central role in regulating intracellular calcium release [76], driving cardiac contraction [77]. Atrial function remains impaired after sinus rhythm restoration in AF patients, primarily due to reduced $I_{CaL}$ [78]. Less intuitive was the improved atrial function following impaired $I_{K1}$. Experimental studies show that increased intracellular calcium reduces $I_{K1}$ [79–81], making it plausible that impaired $I_{K1}$ raises intracellular calcium. To our knowledge, this is the first study delineating the impact of individual fibrosis-associated changes on atrial function through electromechanical simulations on patient-specific geometries. However, our results align with those of Hurtado et al. [82], who found L-type calcium channel conductance, followed by potassium delayed rectifier conductances, to be the most sensitive parameters in ventricular electromechanical model representations. They also found L-type calcium channel

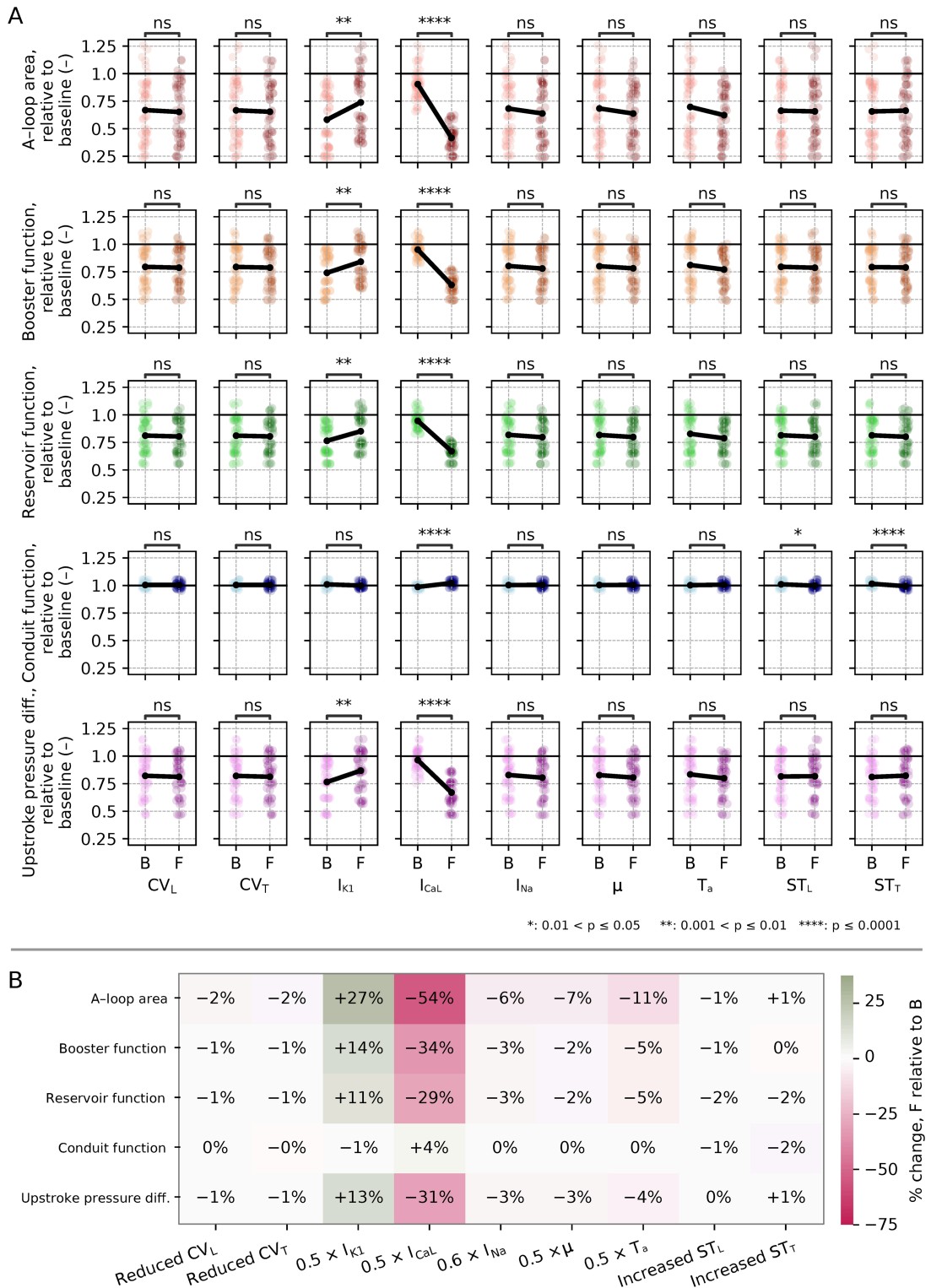

**Fig 11. FFD – main effect, original fibrosis burden.** (A) Main effect plots for each parameter and each metric, across all simulation combinations (Table 3) and all three patients. In each subplot, the combinations were divided into B or F groups based on whether the given parameter was set to baseline or fibrotic value. Points display values from individual simulations (i.e., all FFD combinations for all three patients), with lighter colors for the B group and darker colors for the F group. Lines show average change from B to F groups. (B) Average increase/decrease across all patients and combinations for each metric, calculated as the difference between F and B relative to B levels.

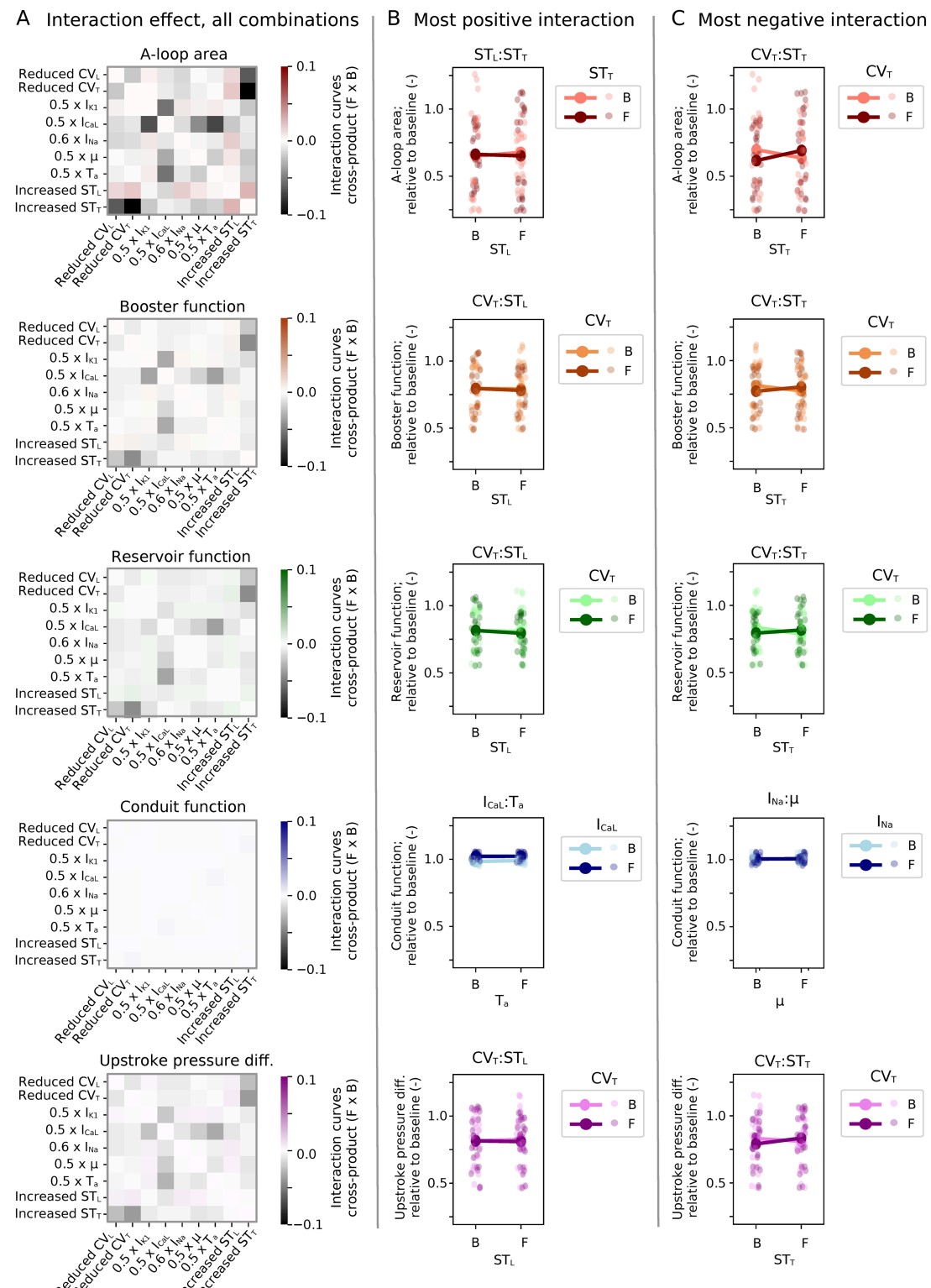

**Fig 12. FFD – interactive effect, original fibrosis burden.** (A) Interactive effects for each parameter pair, as calculated by Eq (6), across all FFD combinations (Table 3) and all three patients. (B–C) Interaction lines for most positive (enhancing, B) and most negative (mitigating, C) impact. Points represent values from individual simulations (i.e., all FFD combinations for all three patients), while lines display average changes within subgroups.

https://doi.org/10.1371/journal.pcbi.g012

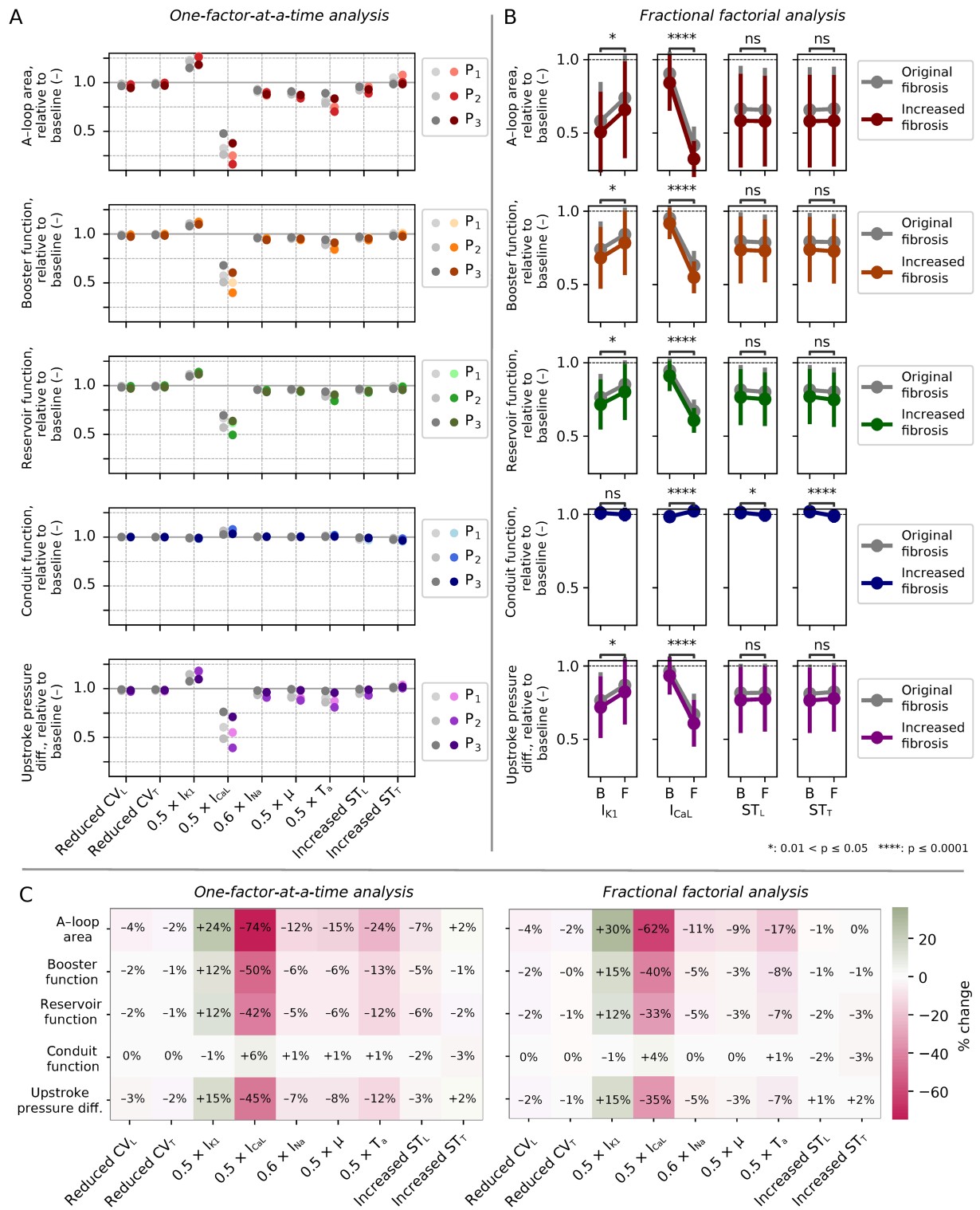

**Fig 13. Impact of 50% synthetically elevated fibrosis: OFAT analysis and FFD analysis, main effect.** (A) Results from the OFAT analysis for Patient 1–3 (P1–P3) patient. Original fibrosis levels are indicated by gray points on the left (the same as displayed in Fig 10) and extended fibrosis on the right. (B) FFD main effect: Error bars indicate standard deviation, and gray plots represent original fibrosis levels (based on the same underlying data as Fig 11). Comparisons for significant differences were performed for elevated fibrosis simulations. Only significant factors are displayed here, while plots for all are included in Fig F in S1 Text as described in Sect D in S1 Text. (C) Percentage impact of each parameter, averaged across all three patient cases, for OFAT (left) and the FFD (right) analyses.

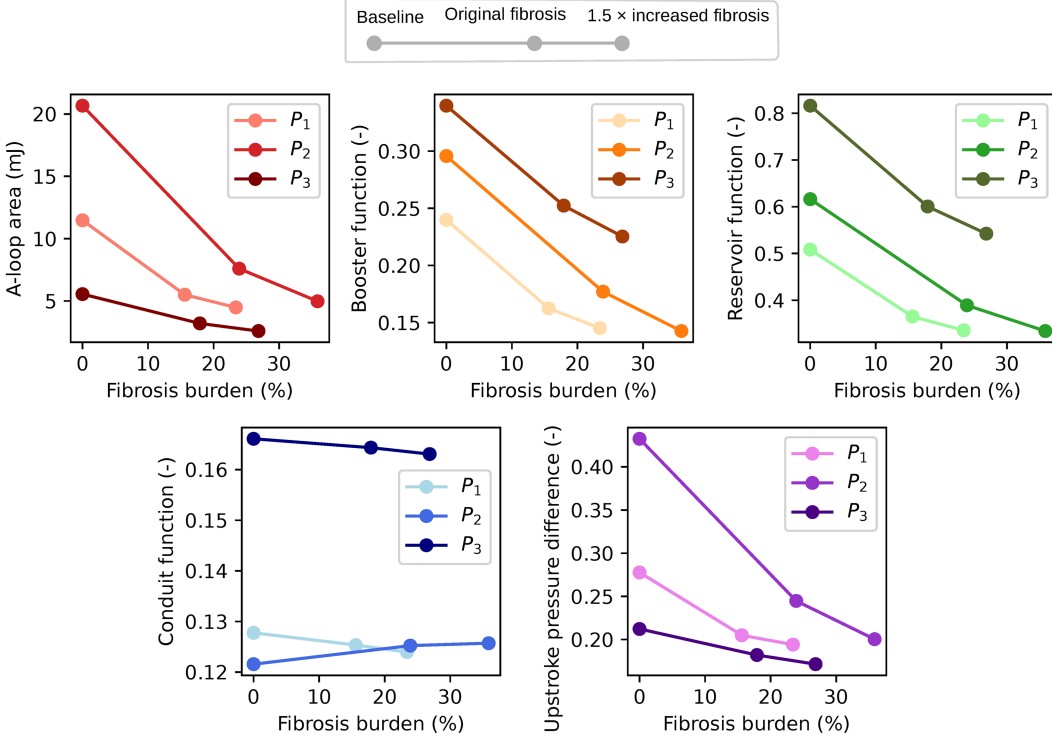

**Fig 14**. **Absolute values of our five metrics for increasing levels of fibrosis, fully fibrotic simulation.** Absolute values of A-loop area (stroke work), booster function, reservoir function, conduit function, and upstroke pressure difference for Patients 1–3 ($P_1$–$P_3$). For each patient, metrics are shown for baseline (0% fibrosis), original, and 50% synthetically increased fibrosis burdens.

conductance raise intracellular calcium amplitude, whereas increasing potassium delayed rectifier conductances reduced it, consistent with our results and experimental work.

Strocchi et al. [83] performed an extensive sensitivity analysis encompassing all four cardiac chambers. Transverse atrial stiffness ($b_{t,A}$) had a stronger effect than L-type calcium channel conductance on end-systolic and end-diastolic LA volumes, whereas potassium channel conductances had little to no impact. The apparent discrepancy with our results may reflect differences in sensitivity analysis output metrics and parameterization. We observed increased stiffness to shift end-systolic and end-diastolic LA volumes but not their relative difference (i.e., booster function). Additionally, the stiffness parameter in Strocchi et al. [83] was an exponent with implied higher sensitivity. They also imposed increased stiffness throughout the atrium, whereas we applied it only to fibrotic regions.

Stiffness has also been explored in other models. Moyer et al. [17] found that fibrosis-associated increases in global stiffness reduced both systolic and diastolic function, with increased pressure and reduced volume for the latter. Meskin et al. [72] reported that higher compliance (lower stiffness) decreased pressure and increased A-loop area. Both studies advanced understanding of LA biomechanics but applied uniform stiffness changes, ignoring heterogeneity introduced by fibrotic remodeling. They also differed in material representation: our model used a reduced Holzapfel-Ogden law with a conditional term (combined with detailed fiber orientations), while the formulation used by Moyer et al. increased fiber-direction stiffness under both contraction and stretch, and Meskin et al. used silicone rubber, which is isotropic. Pericardial boundary conditions in our model may further dampen stiffness effects. We found that increased stiffness in fibrotic regions minimally reduced A-loop area, likely because changes were confined to fibrotic regions, while slightly lowering diastolic pressure, probably due to altered mitral valve flow. Gonzalo et al. [21] observed a larger systolic volume

reduction ( 5% vs. 2%), likely reflecting the fivefold stiffness increase in fibrotic regions, while diastolic volume remained unchanged. Physiologically, non-fibrotic regions are also probably stiffened due to interstitial fibrosis, though substantial stiffness heterogeneity remains. It seems plausible that changes in stiffness governing the largest part of the atrial body – perhaps better assessed via a median rather than mean value – will govern the relative impact on LA function.

A shift in LA volume combined with increased stiffness has been observed in clinical studies [84], although this likely included volumetric expansion. LA stiffness indices derived from clinical flow and strain measurements show predictive power [84–86], but not independently of volumetric indices. Lamy et al. [7] used invasive measurements of pressure to calculate the stiffness index and correlated it with fibrotic burden assessed by LGE. They found significant differences in LA strain metrics, but notably not in pressure. Further clinical, experimental, and computational studies are needed to better understand the relationships between LA stiffness, volume, and pressure changes.

### 4.2 Sensitivity analysis takeaways

We performed two types of sensitivity analysis: a simpler OFAT scheme, exploring the isolated impact of each factor, and a detailed FFD scheme, also accounting for interactive effects. We found that most of the effects could be predicted by the OFAT analysis. However, FFD analysis revealed that the impact of $I_{CaL}$ was partially mitigated when combined with other parameter changes, while the impact of $I_{K1}$ was amplified. In the model, these changes can be linked to shifts in equilibrium values altering action potential and calcium transient morphology (Fig 7). Physiologically, they might be considered compensatory feedback mechanisms.

Sensitivity analysis can be performed in various ways, involving many methodological choices. Advanced techniques like the Morris elementary effects method [82,87,88], Sobol indices [83,89–91], and Bayesian history matching [92,93] involve finer interval sampling, capturing non-linear behavior, interaction effects, and sensitivity regions. However, these benefits come at the cost of running more simulations. These can feasibly be carried out through model simplifications, computational optimization strategies, or surrogate models like Gaussian process emulators [54,90–93].

In our study, we sampled all relevant parameters at two levels: baseline and fibrotic. The FFD scheme presented an efficient methodology allowing us to perform our sensitivity analysis with a detailed multi-scale, multi-physics computational model, based on a manageable number of simulations. It enables direct quantification and assessment of the directionality of parameter effects, which is not provided directly through Sobol analysis nor Bayesian history matching, and only statistically approximated in the Morris elementary effects method. Finally, while OFAT is a rudimentary method that (by definition) misses all interactive effects, this attribute makes it an eminently interpretable approach.

### 4.3 Clinical implications

We found that fibrotic remodeling substantially impaired the cardiac function, mostly explained by impaired $I_{CaL}$ With fibrotic area constituting between 15.6% and 23.9% (for original fibrosis burden), for the fully fibrotic simulation (with all factors set to fibrotic levels), A-loop area, booster function, reservoir function, and the difference in upstroke pressure decreased by 53%, 33%, 30%, and 28%, on average. However, with 50% increased fibrosis burden, we only saw a moderate further impairment. Our spatial analysis revealed substantial dispersive impact from the fibrotic to the non-fibrotic regions. The moderate effect could be explained by the fact that synthetically elevated fibrosis mostly covered larger areas by expansion rather than emerging in new locations. As such, these denser areas would have a less dispersive impact into non-fibrotic myocardium. This might suggest the hypothesis that scattered fibrosis distributions, as opposed to dense distributions, are more consequential for atrial dysfunction.

Two of our three patients (Patients 1 and 2, with LA volume indices 55.5 and 48.3) had enlarged LAs, with end-diastolic volumes and volume indices above normal ranges [94]. Using the Utah classification standard, Patients 1 and 3 had fibrosis burden at level II (5–20%), while Patient 2 was at level III (20–35%). Among AF patients, males have larger atria (but

not higher LA volume indices), whereas females have higher fibrosis burden [95] and larger low-voltage areas [96]. This is the opposite of what we observe for Patient 1 (female, largest LA) and Patient 2 (male, highest fibrosis burden), but not unexpected given inter-individual variability.

Several clinical studies have performed statistical analysis assessing LA strain metrics derived from clinical images. Hopman et al. [97] and Chahine et al. [98] found that reservoir, booster (contractile), and conduit strain were reduced in AF patients compared to healthy controls. In our study, we observed a decrease in booster and reservoir function, but not substantially in conduit function. Our model may not accurately capture changes in conduit function, potentially due to the use of a simplified 0D circulatory model or underestimation of traction force during LV contraction. Moreover, we only imposed changes in fibrotic regions. In reality, interstitial fibrosis in AF patients [5] raises myocardial stiffness everywhere and not only in fibrotic regions. Fibrosis levels have been found to be similar for embolic stroke of undetermined source (ESUS) and AF patients [2,99]. Bashir et al. [100] compared atrial strain metrics for ESUS patients to patients with non-cardioembolic stroke. They found that the same metrics (contractile, reservoir, and conduit) were associated with a higher risk of ESUS occurrence, and a higher risk of later AF detection.

## 4.4 Limitations

Most model parameter values were not patient-specific, and those that were had constraints. In calibrating patient-specific CV, we assumed healthy baseline simulation values not considering regional fibrotic changes. However, CV had low impact among the metrics considered, making this simplification reasonable. With mostly generic parameters, the models reflect general atrial characteristics, and varying patient geometries or fibrosis distributions would likely yield similar functional outcomes. The model could further be improved by personalizing parameters in the 0D circulatory model to match clinical volume and pressure-related measurements [101,102].

Beyond parameter assumptions, the model also lack several dynamic feedback mechanisms that could impact atrial function. It omits electromechanical feedback [103–106], and uses a simplified 0D circulatory model for hemodynamic feedback, which cannot fully capture fluid-structure interactions [107–109]. Fluid-structure interaction models that resolve spatial variations in endocardial pressure might be necessary to resolve secondary interactions involving regional fibrotic changes in myocardial properties. Similarly, the use of a lumped 0D model representing the impact of the other chambers, rather than a full 4-chamber model, might not accurately capture interaction between the different chambers [110]. Including at least the left ventricle could better represent the passive phase of atrial function and ventricular remodeling. However, this would greatly increase computational costs and model complexity. A potential future compromise would be to estimate patient-specific values for the CircAdapt parameters corresponding volumes of each cardiac chamber, whereas our present study used default settings for the sake of simplicity.

LAEF values in our simulations (33.70%, 38.13%, and 44.94% at baseline) were consistently lower than corresponding clinical estimates (39.24%, 51.56%, 58.2%; see Table 1). The discrepancy can be attributed to the passive part of the LAEF, which was underestimated in our model (12.78%, 12.16%, 16.61% vs. reference values of 35–40% [49, 50]), whereas active LAEFs agreed with typical normal values (23.99%, 29.57%, 33.97% vs. 30%). In the future, model improvements can be made to better represent passive function. Such improvements might also increase the sensitivity of the conduit function, which we generally found to be low.

The geometrical assumptions introduce additional limitations. We used a uniform myocardial thickness of 2 mm, although atrial wall thickness varies regionally and between patients [34], see also reported ranges in Table 1. Changes in LA volume, fibrosis burden, and wall thickness are all components of LA remodeling. Relationships between these components remain active research areas, with implications for atrial pump function and AF substrate development [34,111–113]. Additionally, myofiber architecture was specified using a rule-based algorithm. Cardiac computed tomography offers the ability to quantify regional wall thickness and patient-specific myofiber orientation [114]. Incorporating such data could enhance model accuracy and predictive value.

Our sensitivity analysis only considered a subset of all possible parameter combinations, with two levels for each parameter. There might be other influential parameters not considered in our study. Our study is also limited by the scarcity of experimental data on the relative change in each fibrosis-associated parameter. Despite these limitations, we believe the directionality of each parameter is within reasonable physiological range. Future experimental studies might refine exact parameter changes.

We only consider three patients, whereas clinical studies typically involve hundreds of patients. Combined MRI and EAM data collection is logistically challenging, while model simulations are time-consuming. Extending the analysis to a larger cohort would increase confidence in our findings and allow more granular analysis. Moreover, our analysis focuses on fibrotic regions identified by LGE, which is normalized to non-fibrotic myocardium. This approach does not capture interstitial fibrosis, which is also present among AF patients [5] and might underestimate remodeling in LGE-identified non-fibrotic regions.

## 4.5 Future work

This study's findings warrant future research quantifying $I_{CaL}$, $I_{K1}$, and myocardial stiffness. Clinical and experimental studies are needed to validate the results presented here, while computational studies can build on our results. Several antiarrhythmic and rate-control agents used for atrial fibrillation contain $I_{K1}$-blocking agents [115], while blockers of atrial-specific potassium channels (Ca$^{2+}$-activated K+ channels of small conductance, SK channels) have been suggested as future therapeutic drugs [116,117]. These may serve as alternatives to other medications, including but not limited to calcium channel blockers [118]. While primarily assessed in terms of arrhythmogenicity, these drugs also impact intracellular calcium and force generation. However, blocking $I_{K1}$ may trigger ectopic activity due to increased resting membrane potential. Careful investigation is warranted regarding the balance between arrhythmic effects and benefit to atrial biomechanical function. An interesting avenue and extension of our work could be exploring the impact of rhythm control medication on atrial and ventricular function, in both sinus rhythm and subject to re-entry waves mimicking those observed in atrial and ventricular fibrillation.

Sensitivity analysis studies of the fibrotic LA could extend in several directions. Incorporation of additional parameters (e.g., other ion channels or parameters relevant to interstitial fibrosis in non-fibrotic tissue) could reveal other influential mechanisms. Wall deformation predicted by electromechanical simulations could also be integrated with spatially resolved analyses of LA flow and coagulation dynamics to assess how fibrotic remodeling affects downstream outcomes such as thrombosis risk [21,119–121]. In an extended sensitivity analysis, one could include spatiotemporal analysis of blood flow, including parameters associated to atrial morphology [122], blood rheology [123] (e.g., hematocrit and red blood cell aggregation timescale), coagulatory state, or anticoagulation regime [120]. The sensitivity analysis scheme could be extended to ventricular models or whole-heart models, considering the impact of ventricular fibrotic remodeling in comparison or in addition to atrial fibrotic remodeling.

Modeling studies with clinical data can yield mechanistically informative results. Image-based strain analysis can be combined with model-based analysis for the same individuals. Such a study would provide detailed insight into where strain analysis deviates from the model, important for model validation. Another avenue could be to combine modeling with regional analysis of fibrosis distributions [124,125], relating fibrosis burden and variations in spatial patterns to changes in LA function. Computational modeling could also compare LA function after pulmonary vein ablation [126,127], by representing veins as non-conductive, stiffer regions and possibly incorporating post-ablation LGE-MRI. Findings could be related to clinical data showing decreased LA function [71,84,128,129].

## 5 Conclusions

In our study, we used a computational model combined with patient-specific LA geometries to analyze the impact of nine parameters related to fibrotic remodeling. Our sensitivity analysis predicted that impairment of $I_{CaL}$ and $I_{K1}$ were most

consequential in terms of changes in LA function, having respectively decreased and improved effect. Future research focusing on these may greatly improve our understanding of fibrotic remodeling and its effects on atrial function. We found that reduction in $I_{CaL}$ and $I_{K1}$ had a dispersive effect impacting non-fibrotic tissue, and that an increase in fibrosis burden produced a comparably moderate reduction in LA function, potentially related to fibrosis density (scattered versus dense). In the future, modeling frameworks with larger cohorts could better elucidate relationships between fibrosis burden, spatial patterns, and LA function impairment. Modeling efforts could include spatiotemporal analysis of thrombogenic risk subject from fibrotic remodeling, ultimately improving risk assessment and prevention strategies.

## Supporting information

**S1 Text. Appendix.** Supplementary text and figures providing further details of methods, underlying data, and extended results.
(PDF)

**S1 Movie. Spatiotemporal electromechanical distributions, Patient 1.** The movies (one for each patient) display spatiotemporal distributions of membrane voltage, intracellular calcium concentration, active tension, and fiber direction strain; for baseline, reduced $I_{CaL}$, reduced $I_{K1}$, and fully fibrotic simulations.
(M4V)

**S2 Movie. Spatiotemporal electromechanical distributions, Patient 2.**
(M4V)

**S3 Movie. Spatiotemporal electromechanical distributions, Patient 3.**
(M4V)

## Author contributions

**Conceptualization:** Christoph M. Augustin, Patrick Boyle.

**Data curation:** Åshild Telle, Ahmad Kassar, Nadia Chamoun, Romanos Haykal, Tori Hensley, Yaacoub Chahine.

**Formal analysis:** Åshild Telle.

**Funding acquisition:** Juan C. del Álamo, Nazem Akoum, Christoph M. Augustin, Patrick Boyle.

**Investigation:** Åshild Telle.

**Methodology:** Åshild Telle, Christoph M. Augustin, Patrick Boyle.

**Project administration:** Patrick Boyle.

**Resources:** Nazem Akoum, Patrick Boyle.

**Software:** Åshild Telle, Christoph M. Augustin.

**Supervision:** Juan C. del Álamo, Nazem Akoum, Christoph M. Augustin, Patrick Boyle.

**Validation:** Alejandro Gonzalo, Oscar Flores, Juan C. del Álamo, Nazem Akoum, Christoph M. Augustin.

**Visualization:** Åshild Telle.

**Writing – original draft:** Åshild Telle.

**Writing – review & editing:** Åshild Telle, Ahmad Kassar, Romanos Haykal, Alejandro Gonzalo, Tori Hensley, Oscar Flores, Juan C. del Álamo, Nazem Akoum, Christoph M. Augustin, Patrick Boyle.

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
