## [Decision Letter · Decision Letter 0]

15 Aug 2025

PCOMPBIOL-D-25-01254

Systematic computational assessment of atrial function impairment due to fibrotic remodeling in electromechanical properties

PLOS Computational Biology

Dear Dr. Boyle,

Thank you for submitting your manuscript to PLOS Computational Biology. After careful consideration, we feel that it has merit but does not fully meet PLOS Computational Biology's publication criteria as it currently stands. Therefore, we invite you to submit a revised version of the manuscript that addresses the points raised during the review process.

While the reviewers' agreed about the interest of this work for the field, they identified multiple points that have not been fully elucidated in the manuscript and/or put into context with other works in the field.   

Please submit your revised manuscript within 30 days Oct 15 2025 11:59PM. If you will need more time than this to complete your revisions, please reply to this message or contact the journal office at ploscompbiol@plos.org. Please include the following items when submitting your revised manuscript:

We look forward to receiving your revised manuscript.

Kind regards,

Anna Grosberg, Ph.D.

Academic Editor

PLOS Computational Biology

Marc Birtwistle

Section Editor

PLOS Computational Biology

**Journal Requirements:**

**Reviewers' comments:**

Reviewer's Responses to Questions

**Comments to the Authors:**

Reviewer #1: Telle et al. use an electromechanical model to quantify the relative contributions of fibrotic remodeling on left atrial function. They identified two calcium-related properties that explain most the reduction in atrial function following fibrosis. The use of the model to explain why these parameters are consequential for LA function is much appreciated. The study design is sound, manuscript is well-written, and figures support the methods and findings and are clearly labeled. Congratulations on the authors for this insightful work. However, I have several comments that need to be addressed, which hopefully will further strengthen the manuscript.

Major comments:

- The choice of patients could significantly influence the study outcomes, with substantial differences already observed between the baseline properties (Table 1) of the three selected patients. Was there a rationale for choosing these three patients? Were the patients selected from a larger cohort? Please provide a more detailed description of the patient recruitment.

- LA EDV for was around two-fold greater for Patients 1 and 2 compared to Patient 3. Was this because of remodeling or body size? Reporting body surface area (or another related metric) in Table would be helpful for this.

- Related to the previous comment: a constant wall thickness was assumed in all patients. However, would a different wall thickness be expected between Patients 1 and 2 and Patient 3?

- It is not completely clear to me how the 50% fibrotic burden was achieved. Was the threshold lowered until the target total burden was achieved?

- Is a different fiber architecture expected in fibrotic areas? If so, does the rule-based fiber direction method account for this?

- “Active tension generation was scaled with 50 kPa at baseline, calibrated to give approximately 30% emptying fraction”. Was Cine MRI obtained? If so, did these data show a similar emptying fraction, and are any differences between patients observed/expected? It would be helpful to show the data in Fig. 3 (middle row), either full cycle or end-diastolic volume to indicate how well the model was calibrated for each individual patient.

- FFD is an elegant approach to efficiently investigate parameter sensitivity. Recent work (for example: Strocchi et al., Longobardi et al., Jones & Oomen) leveraged surrogate models to perform a global sensitivity analysis on (electro-)mechanical cardiac model. It would be helpful to comment on how FFD differs from global methods like Sobol’s method beyond computational cost.

- Could the reduced-order approach for the other chambers have contributed to the differences in findings compared to Strocchi et al., who used a four-chamber cardiac model?

- While I appreciate the study is not statistically powered to identify potential sex differences, are any sex differences expected in atrial fibrotic remodeling?

Minor comments:

- There is a broken link on line 296.

- What causes the oscillations in the bottom right of the PV loops?

- Fig. 9b: legend’s red line is mislabeled.

Reviewer #2: The manuscript by Telle et al. presents an in silico investigation of atrial electromechanical abnormalities following fibrotic remodeling based on person-specific patterns. Nine parameters were investigated alone or in combination to understand their contribution to different metrics of atrial performance. This is an important problem as number of AF patients increases.

The most striking result was that reduction of IK1 improves atrial performance (increases A-loop and dampens the adverse effects of reduced ICa2+). This result is somewhat paradoxical and requires more in depth discussion. Please elaborate on the mechanism by which IK1 enlarges the A-loop. Reducing IK1 depolarizes the resting membrane potential, which may be pro-arrhythmic by itself, or lead to undesired effects. Blockers of IK1 can indeed reduce excitability and be anti-arrhythmic, but dosing these to the right concentration to not trigger ectopic rhythm is tricky. This finding has to be put into context.

Perhaps another somewhat surprising finding is the very modest effect of stiffness on the metrics for atrial performance in this study. Please comment in more detail how these predictions compare to others and the clinical observations and what difference in assumptions exist between computational studies (some of that is covered in the Discussion yet can be clarified further). What are the limitations of the currently used lumped parameter model of hemodynamics (0D model)?

Other questions and suggestions:

• In the Abstract, would be better to either avoid or define terms (e.g. A-loop) that are not commonly used outside a specific area, for clarity to a broader audience.

• Are the listed conduction velocities correct and are they average values, incl. fibrotic regions? They seem low, unless the units are different.

• Was the fiber direction personalized per patient or generic? Would you expect that to play a role for better prediction?

• In Figure 9B, the legend needs to be corrected for the red line to IK1.

**Have the authors made all data and (if applicable) computational code underlying the findings in their manuscript fully available?**

Reviewer #1: **No: **Part of the software that was used to generate the results is under a commercial license.

Reviewer #2: Yes

PLOS authors have the option to publish the peer review history of their article (what does this mean?). If published, this will include your full peer review and any attached files.

Reviewer #1: **Yes: **Pim J.A. Oomen

Reviewer #2: **Yes: **Emilia Entcheva

**Figure resubmission:**
---

## [Decision Letter · Decision Letter 1]

3 Nov 2025

Dear Prof. Boyle,

We are pleased to inform you that your manuscript 'Systematic computational assessment of atrial function impairment due to fibrotic remodeling in electromechanical properties' has been provisionally accepted for publication in PLOS Computational Biology.

Best regards,

Anna Grosberg, Ph.D.

Academic Editor

PLOS Computational Biology

Marc Birtwistle

Section Editor

PLOS Computational Biology

Reviewer's Responses to Questions

**Comments to the Authors:**

Reviewer #1: The authors have addressed all my comments and concerns.

Reviewer #2: Thank you for a thorough revision.

**Have the authors made all data and (if applicable) computational code underlying the findings in their manuscript fully available?**

Reviewer #1: **No: **Simulations were performed with commercial software and can thus not be made publicly available, all other data is made accessible.

Reviewer #2: Yes

PLOS authors have the option to publish the peer review history of their article (what does this mean?). If published, this will include your full peer review and any attached files.

Reviewer #1: **Yes: **Pim Oomen

Reviewer #2: **Yes: **Emilia Entcheva

---

## [Editor Report · Acceptance letter]

PCOMPBIOL-D-25-01254R1

Systematic computational assessment of atrial function impairment due to fibrotic remodeling in electromechanical properties

Dear Dr Boyle,

I am pleased to inform you that your manuscript has been formally accepted for publication in PLOS Computational Biology. Your manuscript is now with our production department and you will be notified of the publication date in due course.

With kind regards,

Judit Kozma
